# Oxidative Stress: A Suitable Therapeutic Target for Optic Nerve Diseases?

**DOI:** 10.3390/antiox12071465

**Published:** 2023-07-20

**Authors:** Francesco Buonfiglio, Elsa Wilma Böhm, Norbert Pfeiffer, Adrian Gericke

**Affiliations:** Department of Ophthalmology, University Medical Center, Johannes Gutenberg University Mainz, Langenbeckstrasse 1, 55131 Mainz, Germany; elsawilma.boehm@unimedizin-mainz.de (E.W.B.); norbert.pfeiffer@unimedizin-mainz.de (N.P.)

**Keywords:** oxidative stress, optic nerve, retinal ganglion cell, glaucoma, Leber’s hereditary optic neuropathy, ischemic optic neuropathy, optic neuritis

## Abstract

Optic nerve disorders encompass a wide spectrum of conditions characterized by the loss of retinal ganglion cells (RGCs) and subsequent degeneration of the optic nerve. The etiology of these disorders can vary significantly, but emerging research highlights the crucial role of oxidative stress, an imbalance in the redox status characterized by an excess of reactive oxygen species (ROS), in driving cell death through apoptosis, autophagy, and inflammation. This review provides an overview of ROS-related processes underlying four extensively studied optic nerve diseases: glaucoma, Leber’s hereditary optic neuropathy (LHON), anterior ischemic optic neuropathy (AION), and optic neuritis (ON). Furthermore, we present preclinical findings on antioxidants, with the objective of evaluating the potential therapeutic benefits of targeting oxidative stress in the treatment of optic neuropathies.

## 1. Introduction

Optic nerve diseases encompass a wide range of disorders characterized by optic nerve atrophy, resulting from the loss of retinal ganglion cells (RGCs) and leading to sight-threatening conditions [1,2,3]. These pathologies include:Glaucoma: Glaucoma has a worldwide prevalence of approximately 3.54% in the population aged 40–80 years [4]. It is one of the leading causes of irreversible blindness globally [5], along with cataract and age-related macular degeneration [6,7];Hereditary optic neuropathies, such as Leber’s hereditary optic neuropathy (LHON): LHON has a prevalence of 2–4 in 100,000 for complete penetrance [8,9,10,11,12]. In cases of incomplete penetrance, the prevalence can reach 1 in 800 [13,14];Anterior ischemic optic neuropathies (AION): This category includes arteritic forms, like giant cell arteritis (GCA), which has a pooled prevalence of approximately 51.74 in 100,000 for individuals over the age of 50 [15]. Nonarteritic forms have a reported prevalence of approximately 102.87 in 100,000 in the general population over the age of 40 in the Republic of Korea [16];Inflammatory diseases, such as optic neuritis: Optic neuritis has a prevalence of around 115 in 100,000 in the age range of 20–45 years and approximately 115 in 100,000 overall [17,18,19,20];Traumatic optic neuropathies: The overall incidence of traumatic optic neuropathies ranges from 0.7% to 2.5% [21,22,23,24];Dysthyroid optic neuropathies: These occur in 3–7% of individuals with Graves orbitopathy, which itself has a prevalence of 90 to 300 in 100,000 [25,26,27];Infiltrative optic neuropathies, such as leukemic optic neuropathy, which presents in approximately 16% and 18% of all chronic and acute leukemia cases, respectively [28];Congenital anomalies of the optic nerve, such as optic nerve hypoplasia: The estimated prevalence of optic nerve hypoplasia is between approximately 10.9 and 17.3 in 100,000 individuals under the age of 18 [29,30];Nutritional and toxic optic neuropathies: The prevalence of these conditions varies depending on social and geographical factors [31,32].

Glaucoma is the most prevalent optic nerve disease worldwide (Figure 1) [6,7]. LHON has a low estimated prevalence for complete penetrance cases [8,9,10,11,12], while the prevalence is much higher for carriers of mutation variants in the general population [13,14]. The common underlying feature in all optic nerve diseases is the damage and loss of RGCs and their axons, which gradually leads to optic nerve degeneration [3,33,34]. RGCs have high energy requirements and are particularly susceptible to alterations in their energy supply, mainly generated in the mitochondria through the electron transport chain (ETC) [35]. Oxidative stress plays a pivotal role in the pathophysiology of optic nerve diseases such as glaucoma, LHON, and AION. Imbalances between reactive oxygen species (ROS) as well as reactive nitrogen species (RNS) generation and antioxidant systems lead to reactive species overproduction, adenosine triphosphate (ATP) insufficiency, irreversible cellular injuries, and ultimately RGC loss [3,36,37,38,39,40,41,42,43,44,45].

Considering the high global prevalence of glaucoma and the limited treatment options available for most optic neuropathies [33], it becomes essential to explore new research avenues and investigate novel therapeutic approaches. This review aims to shed light on the role of oxidative damage in the pathophysiology of optic nerve diseases. Specifically, we provide an overview of the current understanding of oxidative stress as a key pathological factor and explore its potential as a viable therapeutic target. We will especially focus on four of the most prevalent and clinically challenging optic nerve diseases, glaucoma, LHON, AION, and ON. Ultimately, our goal is to identify novel curative strategies that may pave the way for improved treatment options in these conditions.

## 2. Anatomy and Perfusion of the Visual Pathway

The optic nerve, also known as the second cranial nerve [46], is composed of thin (0.1 µm) and lengthy (~50 mm) RGC axons that extend from the retina to the lateral geniculate nucleus, resulting in a soma/axon ratio of approximatively 1:10,000 [35]. Within the retinal layers, these axons merge to form the retinal nerve fiber layer [47], which runs parallel to the superficial blood vessels. The inner retina, including the outer plexiform layer through the nerve fiber layer, is supplied by the central retinal artery. On the other hand, the avascular outer retina, consisting of the outer nuclear layer and photoreceptors, receives its blood supply through diffusion from the choriocapillaris. The choriocapillaris is nourished by short posterior ciliary arteries, which branch from the ophthalmic artery [48,49]. Notably, retinal oxygenation exhibits variability depending on light or dark conditions due to the different oxygen demand of rods and cones. In humans, rods, which are responsible for vision in low light, are present in approximately 120 million, while cones number around 6 million [50]. Consequently, retinal oxygen consumption is reduced by half in the presence of light, attributed to the decreased activity of rods compared to cones [50].

Optic nerve axons account for approximately 38% of all axons within the central nervous system [51]. Around 1.2 million RGC axons converge to form the optic nerve head (ONH), also referred to as the optic papilla or optic disc. The ONH exhibits a brighter central depression known as the optic cup [51,52,53,54]. Blood supply to the ONH in humans is provided by the arterial circle of Zinn–Haller [48].

The optic nerve can be divided into four compartments: the intraocular segment (1–2 mm), which includes the retinal nerve fiber layer (RNFL) and extends from the ONH to the lamina cribrosa; the intraorbital segment (25–30 mm), which spans from the retrobulbar tract to the optic canal; the intracanalicular segment (5–9 mm); and the intracranial segment (9–10 mm), which extends from the optic canal to the optic chiasm [51]. Four distinct regions can be identified within the optic nerve head: the nerve fiber layer, the prelaminar region, the lamina cribrosa, and the retrolaminar region [47]. The lamina cribrosa serves as a supportive structure for the RGC axons within the ONH [55,56] and consists of approximately 200–300 porous apertures through which the optic nerve passes from the sclera into the retrobulbar cavity [51]. Deformations of the lamina cribrosa may indicate RGC loss and can signify the initial stages of glaucomatous optic neuropathy [57]. Once beyond the lamina cribrosa, the optic nerve becomes myelinated by oligodendrocytes, increasing its diameter from 1–2 mm to 3–4 mm [51]. The intraorbital, intracanalicular, and intracranial segments receive their blood supply from the posterior ciliary arteries as well as the circle of Willis [58].

The two optic nerves converge at the optic chiasm, where nerve fibers originating from the nasal retina of each eye cross over to join the temporal fibers of the contralateral eye [47,59,60]. The blood supply to the chiasm is provided by the circle of Willis [47,52]. From the chiasm, the RGC axons continue their course into the optic tract, which receives perfusion from the posterior communicating and internal carotid artery [52]. Within the optic tract, the nerve fibers undergo rearrangement to align with their corresponding positions in the lateral geniculate nucleus [59]. Fibers carrying visual information from the right visual field project to the left cerebral hemisphere and vice versa [59]. In the lateral geniculate nucleus, the RGC axons synapse with the second-order neurons of the visual pathway, organized in six layers consisting mainly of large and small neurons [47]. Some fibers from the optic tract also synapse with the olivary pretectal nucleus, regulating the pupillary light reflex [52,61]. Additionally, RGC axons containing melanopsin terminate in the suprachiasmatic nucleus, a crucial center for controlling the circadian rhythms [47,61].

The large and small axons of the lateral geniculate nucleus form optic radiation, which initially projects anteriorly and then turns posteriorly, terminating in the occipital lobe where the visual cortex (Brodmann area 17) is located [47,59]. These regions receive blood supply from branches of the internal carotid artery (lateral geniculate nucleus and optic radiation) and the posterior cerebral artery (visual cortex) [52]. In Figure 2, the perfusion and the anatomy of the visual pathway are illustrated.

An interesting aspect of the optic nerve anatomy is the presence of an unmyelinated portion in the RGCs [35]. In the unmyelinated compartment, specifically in the intraocular segment of the optic nerve, potential signals cannot be transmitted through saltatory conduction due to the absence of myelin [62]. To compensate for this limitation and enable rapid transmission, RGCs generate higher quantities of ATP in their axons to repolarize the plasma membrane [63]. Mitochondrial bidirectional transport (antero- and retrograde) along the axons plays a crucial role in this process. These organelles move toward regions with high energy demands, such as the unmyelinated portions, and ATP gradients are believed to guide this transport [35]. This mechanism may explain the specific vulnerability of RGCs to mitochondrial dysfunction, leading to the triggering of ROS production in a vicious cycle [44,63].

In this context, it is important to highlight the trophic role of myelin in the optic nerve sheath. Myelin has been shown to play a vital role in supplying nutrients to the axon, as the entire mitochondrial respiratory chain has been detected in the myelin sheath of the optic nerve [64]. This finding provides possible explanations for the link between myelin loss and axonal degeneration observed in neuropathies or demyelinating disorders [64]. A decrease in myelin-related mitochondrial respiration may be one of the main triggers responsible for neurodegenerative events [65]. With aging, there is a loss of structural integrity in the myelin sheath, which can subsequently lead to axonal deterioration [66]. This process may underlie various neurodegenerative pathologies, including Alzheimer’s disease, as recent studies have suggested [67]. Therefore, an intriguing future therapeutic approach for neurodegenerative disorders involves improving the integrity of the myelin sheath, which could potentially slow disease progression [67]. In this regard, the comanipulation of microglia and a specific signaling pathway, such as the G protein-coupled receptor 17 pathway in oligodendrocyte precursor cells, has been shown to induce robust myelination and promote axonal regeneration following injury [68]. Exploring these avenues may also offer new therapeutic perspectives for other neurodegenerative diseases, including optic neuropathies [69].

## 3. General Mechanisms of Nitro-Oxidative Stress in the Optic Nerve

### 3.1. Generation of Reactive Oxygen and Nitrogen Species

Mitochondria are vital intracellular organelles responsible for essential chemical reactions that produce energy substrates [70,71]. In addition to their various cellular functions, such as modulating intracellular calcium levels, synthesizing nucleotides, lipids, and amino acids, and regulating apoptosis, mitochondria also generate ROS [70,72,73,74]. ROS, at basal levels, serve as critical mediators of signaling pathways, including hypoxic and inflammatory pathways [44,70,75,76]. The fundamental function of mitochondria is to regulate oxygen metabolism and produce energy in the form of ATP [70,71,77]. The electron transport chain (ETC) within the inner mitochondrial membrane plays a central role in this process [78]. Despite the efficiency of oxidative phosphorylation, electron leaks can occur, leading to the direct interaction of electron carriers with molecular oxygen (O_2_) in the mitochondrial matrix. This interaction results in the donation of electrons and the generation of superoxide (O_2_^•−^) [71,77,79,80]. While mitochondria are recognized as the main source of ROS in the cell, other significant sources include the enzymatic activities of nitric oxide synthase (NOS) and nicotinamide adenine dinucleotide phosphate (NADPH) oxidase (NOX) [3,79,81,82]. NOS generates nitric oxide (NO), while NOX (comprising seven isoforms: NOX1, -2, -3, -4, -5, DUOX1, -2) transfers electrons from cytosolic NADPH to molecular O_2_, generating O_2_^•−^ [83,84].

NO is a free radical that plays a pivotal role in various physiological functions [85]. It serves as a regulator of vascular tone [86,87,88]. Additionally, NO acts as a signaling molecule in neurotransmission and as a regulator of gene transcription [89,90,91,92,93,94]. The production of NO is facilitated by the activity of NOS, an enzyme that has three isoforms: neuronal NOS (nNOS or NOS I); inducible NOS (iNOS or NOS II); and endothelial NOS (eNOS or NOS III) [85,95]. NO rapidly and spontaneously reacts with O_2_^•−^ through a “diffusion-limited reaction” [96,97]. As a result, a highly damaging RNS termed peroxynitrite (ONOO^−^) is generated [85,96,97]. Peroxynitrite contributes to the pathogenesis of diverse retinal disorders, being also newly proposed as a critical factor in the pathogenesis of glaucoma [98,99,100].

### 3.2. Oxidative Damage and Antioxidant Defense Systems

ROS and RNS play a physiological role in cellular responses to hypoxia, cell proliferation, cell death, inflammation, or infection [44,76]. Immune cells, such as phagocytes, produce ROS, which provide reactions necessary for an appropriate killing of pathogens [101,102]. Due to endogenous or exogenous trigger factors, the balance between pro- and antioxidant systems can be critically undermined, resulting in nitro-oxidative stress. In this context, radicals begin to compete for paired electrons with intracellular substrates [103], creating oxidative damage. Oxidative injuries are recognized to be a crucial player in the pathogenesis of a variety of pathologies, including ocular diseases [36,104,105,106,107]. At the biomolecular level, three general forms of injuries caused by reactive species can be distinguished: DNA lesions [108,109], protein alterations [110,111], and lipid peroxidation [96,112]. The consequences of DNA damage are modifications in the expression of proteins and the altered regulation of fundamental activities, like oxidative phosphorylation, according to the vicious cycle theory [113,114,115]. In this context, mitochondrial ROS also induce activation of the nod-like receptor family pyrin domain-containing 3 (Nlrp3) inflammasome, a key factor in pyroptotic cell death during inflammation [116].

Antioxidant systems are responsible for defending cells and tissues from the damaging impact of reactive species, which are constantly produced as a “by-product” of oxidative phosphorylation but also serve, at basal levels, physiological functions [76]. Enzymatic antioxidants comprise SOD, catalase (CAT), glutathione peroxidase (GPX), glutathione-S-transferase (GST), heme oxygenase (HO), peroxiredoxin, and thioredoxin [103,117,118,119,120,121,122].

Nonenzymatic antioxidants can be classified into direct and indirect agents. Direct antioxidants react with ROS or RNS, “being sacrificed in the process of their antioxidant actions” [123,124]. Free radical scavengers are, for example, glutathione (GSH) [125], carotenoids [126], vitamin C (ascorbic acid) [127], and vitamin E (α-tocopherol) [128]. Alternatively, indirect antioxidants are molecules, such as vitamin C, that upregulate antioxidant proteins, for example, via the nuclear factor erythroid-2-related factor 2 (Nrf2) [123,127] or molecules, like α-lipoic acid [129]. Examples of antioxidant compounds adsorbed with the food are resveratrol and betulinic acid [98,130,131].

### 3.3. Oxidative Stress in Retinal Ganglion Cells

The retina belongs to the metabolically most active organs in the human body [132] and requires a relatively large amount of energy substrates [133], which makes it particularly vulnerable to energy insufficiency [134]. Oxygen supply is essential for retinal function [135], and its consumption occurs very rapidly, like in the brain [136,137,138]. Hence, conditions that can modify the supply of molecules, such as O_2_, necessary for the production of energy substrates, like ATP, may rapidly generate significant damage in RGCs due to their susceptibility to oxygen deficiency (Figure 3). Thus, an appropriate blood supply via retrobulbar and retinal vessels is crucial for the proper function of RGCs. Studies in retrobulbar blood vessels reported that ROS blunted endothelial function partially by reducing the contribution of the NOS pathway to endothelium-dependent vasodilation [139]. Likewise, moderately elevated IOP induced endothelial dysfunction in retinal arterioles together with RGC loss [140,141]. Zadeh et al. found in apolipoprotein E (ApoE)-deficient mice that hypercholesterolemia caused oxidative stress and endothelial dysfunction in retinal arterioles but did neither lead to increased ROS levels in the RGC layer nor to loss of RGCs, indicative of compensatory effects [142]. In contrast, a study in pigs reported that after only 12 min of ocular ischemia and 20 h of reperfusion, endothelial dysfunction, retinal edema, and RGC loss occurred [143]. ROS generation due to ischemia/reperfusion (I/R) injury is reported to be caused by diverse enzymes involved in the regulation of oxidative metabolism, such as NOX2, xanthine oxidase (XO), uncoupled eNOS, and by ETC dysfunction [143,144,145,146]. Hyperglycemia was also described to be a cause of endothelial dysfunction and oxidative stress in the retina [147,148,149] via the involvement of NOX2 due to the activation of the receptor of an advanced glycation end product (RAGE)-, mitogen-activated protein kinase (MAPK)-, polyol-, protein kinase C (PKC)-, renin–angiotensin system (RAS) signaling pathways [150,151,152,153,154,155].

## 4. Oxidative Stress in Individual Optic Nerve Diseases

### 4.1. Glaucoma

#### 4.1.1. General Aspects

The term “glaucoma” encompasses a group of optic nerve diseases that share similar morphological characteristics but may have diverse origins. They represent the most common optic nerve disorders [34,156,157], characterized by alterations in the ONH, which include a decrease in the neuroretinal rim and an enlargement of the optic cup, leading to deformation of the lamina cribrosa [34,158]. As a result, in the early stages, typical visual field defects, which follow the arcuate pattern corresponding to the arrangement of nerve fiber bundles, can be observed [159]. Elevated intraocular pressure (IOP) is described as the primary risk factor, which can result from pathological resistance to aqueous humor drainage through the trabecular meshwork (TM), causing mechanical stress and compression on the axonal fibers and the ONH [160,161,162]. Glaucomatous disorders are classified into primary forms, where there is no associated ocular disease, and secondary forms, where known coexisting pathological processes (such as uveitis, neovascularization, trauma, or lens-related conditions) lead to IOP elevation and thus to the development of glaucomatous damage [34,160,163]. Primary and secondary forms are further divided into open-angle glaucoma, which among others includes a subtype known as normal tension glaucoma (NTG) and angle-closure glaucoma, in relation to the angle between the iris and cornea in the anterior chamber [34,164,165]. Primary open-angle glaucoma (POAG), the worldwide major form of glaucoma, describes a disorder where, under conditions of an open chamber angle, structural damage to the optic nerve emerge silently, gradually, and chronically [160]. While high IOP values can be a contributing factor in this disorder, it is not always present in POAG. Elevated IOP is defined as a pressure value that exceeds the 97.5th percentile for the population under consideration and is commonly assumed to be a value higher than 21 mmHg [34,163]. However, it is important to note that elevated IOP is recognized as a risk factor but not a diagnostic criterion for glaucoma [160,163]. In fact, approximately 30–90% of POAG cases have IOP values below the 21 mmHg cutoff, which is considered NTG. The prevalence of NTG varies significantly geographically [166]. Additionally, the term “ocular hypertension” (OHT) refers to a condition where elevated IOP is detected without evidence of glaucomatous optic neuropathy [34]. Angle-closure glaucoma is considered an ocular emergency resulting from anatomical contact between the iris and lens. This contact can cause obstruction of the aqueous humor outflow leading to a condition known as pupillary block, present in approximately 90% of cases of primary angle-closure glaucoma (PACG) [165,167]. In this context, a sudden and significant increase in IOP occurs and can reach values as high as 60 to 80 mmHg [164]. PACG is estimated to affect approximately 26% of the glaucoma population and is responsible for about half of the cases of glaucoma-related visual loss worldwide [168,169,170].

The main risk factors for POAG include elevated IOP, advanced age, black race, myopia, and a positive family history [163]. On the other hand, risk factors for PACG include hyperopia, shallow central anterior chamber depth, shallower limbal anterior chamber depth, anteriorly positioned lens, increased lens thickness, small corneal diameter, and steep corneal curvature, as well as high IOP values, advanced age, Asian ethnicity, and female gender [168,171,172]. Elevated IOP is the only identified modifiable risk factor [173], and reducing IOP is currently the only proven method to treat glaucoma effectively [171,174]. IOP has been considered a central factor in both the “mechanical” and “vascular” theories proposed to explain the etiopathogenesis of glaucoma. According to the mechanical theory, elevated IOP leads to the compression and deformation of RGC axons and the optic nerve, ultimately resulting in cell death due to a reduced or blocked axoplasmic flow and a deficit of cellular supply elements [175]. On the other hand, the vascular theory suggests that decreased perfusion to the optic nerve causes ischemia in RGCs, leading to neurodegenerative damage [161]. A reduced blood supply may be a consequence of IOP-related compression or the functional deficits of blood or vessels that supply the laminar regions of the ONH [176]. Elevated IOP can be a result, for example, of pathological resistance in the aqueous humor drainage process through the TM [162].

The primary goal of antiglaucomatous drugs is to lower IOP to a target level, which is an acceptable range individually set to prevent disease progression [177]. Antiglaucomatous drugs are generally administered topically as eye drops and can be categorized into different classes based on their mechanism of action. Prostaglandin analogues, such as latanoprost, increase uveoscleral and trabecular outflow, while β-blockers (e.g., timolol), α_2_-adrenoceptor agonists (e.g., brimonidine), and carbonic anhydrase inhibitors (e.g., dorzolamide) reduce aqueous humor production [173]. Additionally, α_2_- adrenoceptor agonists, like brimonidine, clonidine, and epinephrine, can also increase trabecular outflow, similar to prostaglandins [173]. Miotic agents, such as pilocarpine, widen the chamber angle, and osmotically active drugs, like mannitol, increase water removal from the eye through the systemic circulation when administered intravenously (IV) [173]. Moreover, pilocarpine was reported to provide a neuroprotective effect through the activation of muscarinic receptors [178]. In the case of PACG, the primary treatment is peripheral iridotomy, a laser procedure that creates a full thickness opening in the iris to relieve pupillary blockage [179]. Additionally, osmotic substances, like mannitol, or carbonic anhydrase inhibitors, such as acetazolamide, can be administered intravenously [173].

The economic implications of glaucoma have been extensively studied worldwide, highlighting its significance as a critical social problem. Cost-of-illness studies have shed light on the financial impact of glaucoma. For instance, a study conducted in the UK in 2002 reported that over £300 million was spent on glaucoma-related expenses [180]. Similarly, an Australian review analyzed published randomized trials and population-based studies since 1985, projecting that the total costs (including direct, indirect, and costs related to the loss of well-being) for POAG in Australia would increase from AUD 1.9 billion in 2005 to AUD 4.3 billion in 2025 [181]. Examining the indirect costs of glaucoma, a study from the UK investigated whether glaucoma can be a risk factor for falls. The study found that between 2012 and 2018, 11.7% of hospital admissions for falls in a national health service hospital were in patients with a secondary diagnosis of glaucoma [182]. Furthermore, a recent retrospective cohort conducted in the US, extrapolating data from 2015 to 2017 and based on a database of patients with POAG and ocular hypertension (OHT), compared these two conditions. The study revealed that advanced stage POAG was associated with a higher risk of falls compared to OHT. Additionally, it was found that the annual eye-related outpatient costs for POAG patients were higher (median: USD 516) than for OHT patients (median: USD 344). Among patients with POAG, those in advanced stages exhibited even higher annual eye-related costs (median: USD 639) compared to those in moderate (median: USD 546) and mild (median: USD 476) stages [183]. Similarly, a previous multicenter study conducted in Germany between 2009 and 2010, involving 2 university hospitals and 13 ophthalmological practices, reported that direct costs for therapies were higher for glaucoma compared to OHT and were directly associated with disease progression. The study highlighted that additional treatment changes necessitated by uncontrolled IOP were the major contributors to the increased costs associated with glaucoma. The study concluded that by effectively managing IOP over the long term and avoiding advanced disease stages, both disease progression and associated costs could be reduced [184]. In summary, publications on this topic consistently emphasize that early first-line therapies play a vital role in reducing the economic burden of glaucoma. By preventing disease progression to advanced stages, these early interventions can help minimize the need for more expensive and sometimes less effective treatments, ultimately alleviating the economic strain imposed by glaucoma [185].

#### 4.1.2. Redox Parameters and Oxidative Stress Biomarkers in Glaucoma

Numerous studies have been conducted to gather evidence of altered redox status in glaucoma by examining oxidant or antioxidant levels in various samples, including blood, plasma, serum, aqueous humor, and the TM. Common systemic parameters used to assess oxidative stress include total antioxidant status (TAS), total antioxidant capacity (TAC), biological antioxidant potential (BAP), total reactive antioxidant potential (TRAP), and total oxidant status (TOS). Biomarkers of oxidative stress, such as 8-hydroxy-2′-deoxyguanosine (8-OHdG), poly-(adenosine diphosphate-ribose)-polymerase (PARP1), oxoguanine-DNA-glycosylase (OGG1), and malondialdehyde (MDA), are indicative of DNA oxidative damage, base excision repair activity, and lipid peroxidation.

An Indian study examining redox-related DNA damage in POAG patients reported elevated levels of 8-OHdG levels in both plasma and aqueous humor. They also observed a negative correlation between 8-OHdG levels and PARP1 and OGG1 levels, which were concurrently lower [186]. Increased levels of 8-OHdG were also found in the TM of patients with POAG, correlating with more severe visual field defects [187,188].

TAC levels were found to be lower in the serum and aqueous humor of glaucoma patients, while MDA levels were increased [189]. Decreased levels of TAS [190] and TRAP [179] were also observed in the aqueous humor of glaucoma patients. Lower TAS levels were additionally found in the plasma samples of individuals with POAG, further associated with an increasing cup-to-disc ratio, which indicates cup–disc enlargement and retinal ganglion cell loss [191]. Studies on serum samples of glaucoma patients reported lower BAP levels, correlating with lower RGC density in young males [192], more advanced visual field loss [193], or higher IOP [194].

A meta-analysis on the redox parameters in glaucoma revealed a general increase in oxidative stress parameters in both serum and aqueous humor, with MDA being the most significant biomarker, suggesting its potential clinical utility [195]. Antioxidative markers were found to be lower in serum, while aqueous humor showed an increase in antioxidant defense, indicating a possible compensatory response to oxidative stress [195]. Similarly, a recent meta-analysis on oxidative stress markers in glaucoma highlighted lower TAS levels in blood and elevated levels of antioxidant enzymes, such as SOD, CAT, and GPX, in the aqueous humor [196]. This systematic review also compared patients with POAG to those with exfoliation glaucoma, the most common secondary form of POAG, characterized by the deposition of fibrillar material in the anterior segment [197,198]. It reported no increase in antioxidant defense in the aqueous humor of exfoliation glaucoma patients, along with a decrease in TAS levels in the blood [196].

RNS, including NO, were found to be increased in the aqueous humor of patients with POAG and PACG [199]. A Spanish study examining both the TM and the aqueous humor of POAG patients reported an upregulation in iNOS and downregulation in eNOS in trabecular meshwork cells (TMCs). They also observed increased levels of MDA in the aqueous humor, corresponding to an increase in visual field defects [200]. The authors of this study suggested that the increased production of NO induced by iNOS may play a role in the process of RGC death, potentially leading to an elevation in MDA levels in the aqueous humor [200].

#### 4.1.3. Oxidative Stress in the Pathogenesis of Glaucoma

To gain a better understanding on the role of reactive species in glaucoma, it is important to discuss the anatomy and function of the TM and the detrimental impact of ROS on these structures. The TM is located in the sclerocorneal angle and consists of three layers: the uveal TM, the corneoscleral TM, and the juxtacanalicular TM, also known as the cribriform TM region. This region is adjacent to the Schlemm’s canal, which allows for the drainage of aqueous humor into the episcleral veins [201]. The TM plays a crucial role in regulating IOP by controlling the outflow of aqueous humor. It allows for the turnover of aqueous humor through drainage into superficial veins toward the conjunctiva [201]. Any alterations in TM permeability, such as changes in TM structure, can result in increased resistance to aqueous humor outflow, leading to elevated IOP. Studies focusing on TMCs have investigated how increased IOP may develop in glaucoma patients. These studies have shown that the oxidative balance in aqueous humor, which has been found to be altered in individuals with glaucoma [179,202], can impact the structure and function of TMCs, particularly the TM endothelial cells lining the Schlemm’s canal [203]. Elevated levels of ROS in the aqueous humor can trigger changes in the TM [162]. This can occur through the fusion of TMCs, leading to trabecular thickening [204]. Additionally, ROS can induce TMC apoptosis, resulting in TM disruption [203]. Alterations in the extracellular matrix (ECM) can also occur, affecting the adhesion of endothelial TMCs to the ECM and causing the TM to collapse [205]. In all of the cases, increased resistance to aqueous humor circulation occurs, leading to elevated IOP [203]. In Figure 4, an anatomical overview of the described organs, with focus on the anterior segment of the eye, is presented.

Oxidative damage contributes to axonal injuries, leading to the death of RGCs [37]. RGC loss may occur as a result of a process involving ROS, elevated IOP, mechanical compression on RGC axons, or vascular compression of optic-nerve-perfusing vessels [40]. Our previous research in mice has demonstrated that already moderately elevated IOP can impair vascular autoregulation and cause endothelial dysfunction in the retina, which is associated with an upregulation in NOX2 [140]. Additionally, high hydrostatic pressure and ischemia have been found to stimulate the release of tumor necrosis factor α (TNF-α) from glial cells, activating apoptotic signaling pathways in RGCs [206]. TNF-α released by active astrocytes, microglia, and Müller glial cells is one of the most significant proinflammatory cytokines in glial neuroinflammation, mitochondrial dysfunction, and oxidative stress [207]. Moreover, TNF-α can induce apoptosis through caspase-8 activation [207]. Mitochondrial dysfunction also plays a pivotal role in glaucomatous optic disorders. Increased IOP has been shown to damage mitochondria, leading to mitochondrial fission and alterations in the expression of the OPA1 gene [208,209]. Via ROS-induced activation of the transcription factor nuclear factor “kappa-light-chain-enhancer” of activated B-cells (NF-kB), mitochondria indirectly contribute to the initiation or amplification of glial neuroinflammation processes [208], resulting in the production of inflammatory cytokines. Furthermore, oxidized mitochondrial DNA and mitochondrial fragments released from microglia can activate inflammasomes [208]. Oxidative stress also disrupts the glutamate/glutamine metabolism, leading to the neurotoxic extracellular accumulation of glutamate [40]. Glutamate is released from dying RGCs and activated glia, and the dysfunctional glial cells are unable to adequately buffer excessive extracellular glutamate [40,210,211]. Additionally, oxidized substrates, such as advanced glycation end products (AGEs) and oxidized low density lipoproteins (oxLDLs) can act as “antigenic stimuli”, inducing ROS production, triggering NF-kB activation, glial activation, neuroinflammation, and apoptosis [40,208]. In this context, the overproduction of ROS activates the apoptosis signal-regulating kinase 1 (ASK-1) [212,213], which, in turn, activates the p38/c-Jun N-terminal kinase (JNK)/extracellular signal-regulated kinase (ERK) axis. This axis triggers the mitochondrial apoptotic pathway, characterized by the release of the Bcl-2-associated X-protein (Bax) and cytochrome c into the cytosol, followed by apoptosome formation (cytochrome c/apoptotic protease-activating factor 1(Apaf-1)/caspase-9), ultimately leading to caspase-3 activation [212,214,215]. Moreover, hydrogen peroxide has been described to activate the phosphoinositide 3-kinase (PI3K)/Ak strain transforming (Akt) axis and reduce the intracellular concentration of the mammalian target of rapamycin (mTor), promoting NF-kB activation [216].

Apart from elevated IOP, other potential risk factors have been investigated in the context of NTG. The “Collaborative Normal-Tension Glaucoma Study” (1998) examined the effectiveness of IOP-lowering therapy in patients with NTG and concluded that although reducing IOP can have a positive impact, it cannot prevent disease progression [217,218]. Recent reviews on NTG have focused on the multifactorial nature of the disease, considering various risk factors implicated in its pathogenesis, including translaminar cribrosa pressure difference, fractal dimensions, neurovascular coupling, vascular dysregulation, endothelial dysfunction, ocular perfusion pressure, and oxidative stress [166,218]. Regarding vascular pathological aspects, Leung and Tham described NTG as part of a larger group of disorders known as small vessel diseases [166]. They focused on the correlation between NTG and cerebral silent infarcts detected in patients with NTG [219,220,221]. CSI can lead to hypoperfusion of the optic nerve head and hypoperfusion in the optic nerve head, increasing the risk of NTG [166]. Additionally, arterial hypotonia, especially during night time when blood pressure is reduced, may result in lower ocular perfusion pressure and decreased blood supply to the ONH [218]. In this context, it is relevant to consider another potential primary injury to RGCs independent of elevated IOP: tissue hypoxia, which can be associated with pathogenic mechanisms in glaucoma [37]. During hypoxia, hypoxia-inducible factor 1-alpha (HIF-1α) induces the transcription of various genes, including those encoding the vascular endothelial growth factor (VEGF), heme oxygenase-1 (HO-1), and inducible nitric oxide synthase (iNOS), aiming to increase oxygen supply to the affected tissue [222,223]. However, HIF-1α also upregulates NOX, which generates ROS, creating a vicious cycle of HIF-1α overexpression and ROS production [224,225]. The presence of increased HIF-1α expression regions in the retina and optic nerve of glaucoma patients was confirmed by Tezel and Wax, further supporting the pathogenic role of hypoxia in glaucoma [226]. Hypoxic conditions primarily cause energy depletion and disrupt ionic homeostasis, leading to increased ROS generation and inflammation [37,227], mediated by the overproduction of ROS via NOX2 [228] and other sources, such as XO and cyclooxygenase (COX) [229]. Additionally, activated glia release TNF-α, which, among other effects, activates NF-kB, intensifying glial activation, neuroinflammation, and apoptosis through caspase-mediated or caspase-independent signaling pathways [207,230].

Reviewing the literature on the pathogenesis of glaucoma (Figure 5), it becomes evident that the pathways leading to RGC loss are redundant, intersect, and overlap, making it challenging to identify a single initial cause-and-effect factor [40]. Ultimately, the outcome appears to be a combination of “vicious cycles”, in which inflammation and ROS interconnect and amplify each other. 

#### 4.1.4. Interplay between Mitochondrial and ER Stress in TMCs and RGCs

Mechanical and vascular stress in glaucoma contribute to mitochondrial dysfunction [210], which leads to calcium imbalance and interconnects with endoplasmic reticulum (ER) stress, resulting in energy impairment and subsequent ROS generation. Numerous studies have explored the relationship between the endoplasmic reticulum and mitochondria in the pathogenesis of glaucoma. Understanding the main processes of this connection can help trace intracellular pathways that contribute to TM dysfunction and RGC death.

The endoplasmic reticulum is an intracellular organelle with multiple functions, including lipid synthesis, calcium storage, and protein processing. It regulates protein folding to ensure proper functionality, facilitates protein transportation, and detects misfolded proteins, which are then retained in the ER for degradation [231,232]. Various conditions, such as hypoxia, oxidative stress, viral infections, nutrient depletion, protein mutations, impaired glycosylation, or disrupted disulfide bond formation, can interfere with the ER’s physiological functions. These disturbances can lead to ER saturation and the accumulation of misfolded proteins in the ER lumen, resulting in endoplasmic reticulum stress [233,234,235]. ER stress activates multiple signaling pathways aimed at restoring cellular homeostasis [231,234]. The unfolded protein response (UPR) is triggered in response to ER stress and encompasses a range of signaling mechanisms designed to reduce protein synthesis, enhance protein folding, and increase protein degradation [236]. Additionally, the ER-associated degradation (ERAD) system is responsible for retro-transporting misfolded proteins from the ER lumen to the cytosol for clearance through the ubiquitin-proteasome system [234,237]. The UPR and ERAD are two independent quality-control mechanisms that can interact to eliminate misfolded proteins and maintain protein folding homeostasis [237]. The UPR can lead to either cell survival or apoptosis [235]. Prolonged activation of the UPR pathways can shift the balance toward cell death [234]. Three main UPR regulators have been extensively described: activating transcription factor (ATF)-6, inositol-requiring protein 1 (IRE1), and protein kinase RNA-like endoplasmic reticulum kinase (PERK). These transmembrane proteins with ER lumen domains serve as sensors of ER stress [235]. ATF-6 activates cascades to enhance the ER folding capacity, promote clearance of misfolded proteins through ERAD, and may also have a proapoptotic effect [231,235,238]. IRE1 initiates a pathway involving spliced X-box binding protein-1 (sXBP1), which promotes ERAD components, ER folding proteins, and autophagy [231]. Furthermore, IRE1 activates JNK through tumor necrosis factor receptor 2 (TNRF-2) and ASK1, leading to apoptosis [235], as well as inflammation via NF-kB induction [239]. PERK activates eukaryotic initiation factor 2α (eIF2α), which downregulates overall protein translation, indirectly protecting the cell from protein misfolding [235]. PERK also upregulates activating transcription factor 4 (ATF-4), which triggers the CCAAT/enhancer-binding protein (C/EBP) homologous protein (CHOP), exerting a proapoptotic function [240]. In this regard, Han et al. demonstrated that a sustained activation of ATF-4 and CHOP increases protein synthesis and leads to cell death through oxidative stress and ATP depletion [236]. These findings align with previous research showing that CHOP deletion in multiple mouse models of diabetes reduces oxidative stress [241]. A noteworthy aspect regarding the connection between ER stress and ROS generation is the ER overload response, which is activated when there is a high concentration of misfolded proteins in the ER lumen [234]. During the ER overload response, a significant amount of calcium ions (Ca^2+^) may be released from the ER, possibly through Ca^2+^ release channels, such as inositol 1,4,5-trisphosphate receptor (IP3R) or ryanodine receptor (RyR) [231,234]. This process can result in increased Ca^2+^ uptake from the ER to mitochondria, leading to the abnormal production of H_2_O_2_ and disruption of the ETC, ultimately causing mitochondrial dysfunction [231,242]. Moreover, UPR-related signaling can activate the endoplasmic reticulum oxidoreductin 1 (ERO1) and NOX, which are involved in oxidative protein folding under normal physiological conditions. However, in the context of ER stress, their activation can contribute to ROS production in the stressed ER [231,243]. Lastly, ROS generation resulting from both ER and mitochondrial dysfunction can activate NF-kB, a key transcription factor involved in inflammation and cell proliferation [244]. The main transductions in the interconnection between ER and mitochondria are summarized in Figure 6.

ER-related oxidative stress has been reported in both TMCs and RGCs in the context of glaucomatous optic neuropathy. Studies on TMCs have demonstrated the involvement of the PERK-eIF2α-ATF4-CHOP cascade in glaucomatous TM, both in human and murine cells, highlighting the activation of this pathway in glaucoma [245,246,247,248]. Similarly, various studies have described the implication of the PERK-eIF2α-ATF4-CHOP pathway in RGC loss [249,250,251,252]. The recent literature has highlighted the potential of targeting the PERK-eIF2α-ATF4-CHOP pathway as a therapeutic approach to prevent CHOP-related oxidative stress and apoptosis, thereby mitigating the TM structure and function loss and potentially reducing IOP elevation. Notably, in a recent study conducted by Gao et al., the protective effect of valdecoxib, a selective COX-2 inhibitor commonly used in the treatment of conditions such as osteoarthritis and rheumatoid arthritis was assessed against apoptosis induced by ER stress. The study demonstrated that valdecoxib inhibits the ATF4-CHOP pathway in “I/R-induced glaucoma-like” damaged cells, providing potential insights into its therapeutic efficacy in glaucoma management [253].

### 4.2. Therapeutic Perspectives in Glaucoma

#### 4.2.1. Therapeutic Potential of Natural Compounds in Glaucoma

Naturally occurring antioxidant compounds offer a diverse range of potential therapeutic options. One such compound is resveratrol, a polyphenol found in peanuts, berries, grapes, and red wine, known for its anti-inflammatory and antioxidant properties in conditions like cancer, neurodegeneration, and aging [254]. In human glaucomatous TMCs, resveratrol has been reported to reduce the expression of proinflammatory molecules, such as IL-1α and iNOS, while increasing the production of NO by elevating eNOS expression. These effects contribute to its beneficial antioxidant effects in the TM [255]. Resveratrol is also an activator of suirtin1 (SIRT1), a nuclear NAD^+^-dependent deacetylase, highly relevant for the regulation of several antioxidant genes, performed by triggering the Nrf2/ARE (antioxidant response elements) pathway [256,257]. In an experimental rat model of glaucoma, resveratrol was shown to delay the loss of RGCs [258]. Additionally, studies by Ye and Meng have demonstrated that resveratrol protects RGCs from H_2_O_2_-induced apoptosis by inhibiting proteins involved in MAPK cascades (p38, JNK, and ERK) and activating antioxidant enzymes such as SOD, CAT, and GSH [259].

Another naturally occurring antioxidant molecule with promising characteristics is curcumin. In chronic high IOP rat models, curcumin has been shown to decrease ROS formation and prevent activation of apoptotic pathways by downregulating caspase-3, Bax, and cyt c [260]. Moreover, in an ex vivo mouse model of optic nerve cut, curcumin has been found to preserve RGC survival by preventing MAPK activation and inhibiting caspase-9 and caspase-3 activation [261].

Spermidine, a natural polyamine present in mushrooms and soybeans, possesses antioxidant properties and has demonstrated the ability to improve RGC loss and visual impairment in murine NTG models [262]. The same study also observed that spermidine treatment in murine optic nerve injury models promoted RGC survival by suppressing the ASK-1/p38/MAPK apoptotic pathway and inhibiting iNOS, particularly in microglial cells [263].

Flavonoids, known for their free radical scavenging properties, have been extensively studied. Extracts from the *Gingko biloba* L. plant, which contain over 70 different flavonoids, have shown interactions with apoptotic pathway proteins such as p53, Bax, Bcl-2, caspase-3, and caspase-9 [264]. These interactions suggest that *Gingko biloba* extracts may reduce RGC damage in glaucoma by inhibiting H_2_O_2_-related apoptosis through pathways involving p53, Bax/Bcl-2, and caspase-3/9 [265]. A dedicated clinical trial evaluated the use of oral antioxidants for glaucoma, comparing extracts of *Gingko biloba* with α-tocopherol for 3 months (NCT01544192), and completed phase III, reporting no clear clinical benefit for the use of *Gingko biloba* [240]. Another flavonoid, coenzyme Q_10_, a natural lipophilic compound, has demonstrated in murine glaucoma models its ability to reduce glutamate excitotoxicity and oxidative stress, promoting RGC survival and preventing apoptotic cell death by decreasing Bax expression and increasing the Bcl-2 associated agonist of cell death (Bad) protein expression [266]. Currently, a clinical trial (NCT03611530) is underway to evaluate the effect of a medical solution containing coenzyme Q_10_ and vitamin E on patients with primary open-angle glaucoma (POAG) [267].

The α-lipoic acid (ALA) is a naturally occurring antioxidant molecule found in various fruits and vegetables, as well as in the heart or liver of animals [268]. In murine glaucoma models (DBA/2J), Inman et al. demonstrated that ALA reduces oxidative stress and upregulates antioxidant agents such as HO-1 and NOS, possibly through the activation of Nrf2 [268]. In a more recent prospective case-control study, an ALA-based solution (also containing taurine, vitamins C and E, lutein, zeaxanthin, zinc, copper, and docosahexaenoic acid) was found to increase TAS and to decrease MDA, a marker of lipid peroxidation, in the plasma of patients with POAG [269].

Additionally, vitamin B3, also known as niacin, has been studied as a potential new treatment for glaucoma due to its antioxidant properties [270]. A study conducted in Korea revealed that patients with NTG have a lower intake of niacin compared to other nutrients, suggesting a possible association between vitamin B3 deficiency and NTG risk [271]. Williams et al. administrated nicotinamide (an amide form of niacin) to DBA/2J mice and demonstrated both preventive and therapeutic effects against the development of glaucoma, preserving the age-dependent reduction in nicotinamide adenine dinucleotide (NAD), a crucial molecule for mitochondrial health [272]. In a recent randomized controlled trial involving 57 patients with glaucoma, nicotinamide supplementation was shown to improve inner retinal function [273].

Currently, there is an ongoing randomized controlled trial (NCT04784234) evaluating a mixture of curcumin, *Ginkgo biloba* extract, alpha-lipoic acid, coenzyme Q10, and other naturally occurring compounds in 100 patients with POAG. The expected completion date for this study is the end of 2023.

#### 4.2.2. NOX Inhibitors

A novel and promising class of antioxidant medications for glaucoma are NOX inhibitors. These compounds offer a new strategy to counteract glaucomatous damage by preserving RGCs from the detrimental effects of neuroinflammation and glial activation, potentially complementing traditional IOP-lowering therapies [274]. Among these compounds, GKT137831, also known as setanaxib, is particularly noteworthy. It acts as a dual inhibitor of NOX1 and NOX4 and has demonstrated protective effects against retinal inflammation and ischemia by reducing hypoxia-induced ROS generation [275]. Another intriguing molecule in this class is GLX7013114, a specific NOX4 inhibitor. In a study involving rats with α-amino-3-hydroxy-5-methyl-4-isoxazolepropionic acid (AMPA)-induced retinal excitotoxicity, intravitreal injections of GLX7013114 were found to attenuate glial activation [276]. Moreover, a recent publication reviewed the role of NOX and NOX inhibitors in glaucoma, highlighting the significant relationship between NOX4 and TGF-β in the fibrotic changes observed in glaucomatous TMCs, which contribute to functional impairment and elevated IOP [274,277]. These NOX inhibitors offer exciting prospects in antioxidant therapy for optic nerve diseases, as they target underlying mechanisms beyond IOP regulation, aiming to protect RGCs and to mitigate neuroinflammatory processes. Continued research in this field holds promise for the development of innovative treatments for glaucoma.

#### 4.2.3. Exploring Nrf2 Activation for Antioxidant Therapy in Glaucoma

Exciting new prospects in antioxidant therapy for glaucoma involve the use of Nrf2 activators, which have the potential to provide benefits through their antioxidant and anti-inflammatory properties. One example is trimetazidine, an anti-ischemic medication that has been shown to activate the Nrf2-HO-1 pathway, leading to the inhibition of RGC apoptosis [278]. Astaxanthin (AST), a potent natural antioxidant found in microalgae and seafood, such as lobster, is another noteworthy compound in this context [279,280]. In a rat model of elevated IOP, AST has been demonstrated to reduce apoptotic pathways [281]. In mouse models of NTG, AST administration inhibited RGC degeneration and suppressed RGC loss [282]. Li et al. showed that AST activates Nrf2 and HO-1 in RGCs of mouse models, resulting in a decrease in RGC loss in glaucoma [283]. Sestrin2, a stress-induced protein with antioxidant properties, has also shown promise in activating Nrf2 by inhibiting the kelch-like ECH-associated protein 1 (Keap-1), thus protecting against RGC apoptosis in mouse tissues [284]. Additionally, eye drops containing metformin, an antidiabetic drug, have been demonstrated to preserve against fibrosis following glaucoma filtration surgery in rodent models by activating the AMP-activated protein kinase (AMPK)/Nrf2 signaling pathway [285]. A recent study in mice demonstrated that sustained intraocular release of erythropoietin can counteract glaucomatous pathogenic processes by reducing superoxide levels in the retina, upregulating antioxidant agents, and activating the Nrf2/ARE pathway through MAPK signaling [286].

#### 4.2.4. Rho Kinase Inhibitors in Glaucoma Treatment

The class of Rho-associated protein kinase (ROCK) inhibitors has shown promise in glaucoma treatment. One notable ROCK inhibitor is netarsudil, which has been demonstrated to reduce fibrosis in the TM, leading to improved aqueous humor outflow and lowered IOP [287]. It has received clinical approval for use in the United States (2017) and Europe (2019) as a 0.02% ophthalmic solution for once-daily topical application [288]. Another ROCK inhibitor, Y-27632, has been shown to upregulate antioxidant enzymes such as CAT and to partially reduce ROS generation [289]. In addition to its beneficial effects on the TM, Y-27632 promotes phagocytosis in glaucomatous TM cells, leading to IOP reduction [290]. Ripasudil, also known as Rho kinase inhibitor K-115, has been evaluated in porcine retinal arterioles and shown to induce endothelium-independent relaxation and inhibit endothelin-1 activity. These findings highlight its potential as a potential antiglaucomatous medication [291]. The development of Rho kinase inhibitors offers a promising avenue for glaucoma treatment. By targeting the ROCK pathway, these inhibitors improve TM function, reduce fibrosis, and potentially lower IOP. Continued research and clinical investigations will further elucidate their efficacy and safety for glaucoma patients.

#### 4.2.5. ER Stress Antagonists: Potential Antioxidant Drugs

Another subgroup of potential antioxidant drug includes compounds that antagonize ER stress. Among these compounds, 4-phenylbutyric acid (4-PBA) has been the subject of several studies. Originally used in urea cycle disorders and later in the treatment of cystic fibrosis in the 1990s [292,293], phenylbutyrate has been reported to attenuate ROS production in activated microglia [294]. Additionally, 4-PBA has been found to reduce oxidative stress caused by a high-fat diet or acute ammonia challenge by counteracting ER stress [295]. Furthermore, 4-PBA has shown the ability to decrease ER stress and prevent disease phenotypes in murine glaucoma models [296]. Importantly, it has also been demonstrated to lower IOP by promoting ECM degradation through the activation of matrix metalloproteinase (MMP)-9 [297].

#### 4.2.6. Other Antioxidant Molecules

In addition to the previously mentioned compounds, several other antioxidant molecules have shown potential therapeutic effects in glaucoma.

For example, edaravone, a medication primarily used against stroke and known for its free radical scavenging properties [298] was described to suppress pJNK/p38 proapoptotic pathways in glaucoma models [43], preserving RGCs from death [299,300].

Valproic acid (VPA), a widely used antiepileptic drug, has been shown to upregulate antioxidant enzymes, such as SOD, CAT, and GPX, while inhibiting apoptotic pathways in rodent models of retinal ischemia–reperfusion injuries [301]. In murine models of NTG, VPA has been found to decrease oxidative stress and enhance RGC survival through a pathway associated with ERK [302]. Additionally, a recent Swedish study using retina explant models suggested that VPA possesses anti-inflammatory properties by reducing the expression of proinflammatory cytokines and attenuating microglial changes, highlighting its potential as an antineuroinflammatory drug in retinal diseases [303]. A randomized controlled trial assessing the effectiveness of VPA in glaucoma patients reported an improvement in visual acuity among patients with advanced glaucoma [304].

N-acetylcysteine, a molecule with radical scavenger properties originally used as antidote in cases of paracetamol overdose and more recently, as a mucolytic agent in bronchopulmonary disorders [305] has shown promise in mitigating retinal oxidative stress caused by high IOP when combined with brimonidine therapy in rat models of OHT [306]. It has also been demonstrated to target the HIF-1α pathway via BNIP3 (Bcl2 interacting protein 3) and the PI3K/Akt/mTOR pathway, thereby preventing hypoxia–mimetic induced autophagy in RGCs [307]. In murine models of NTG, N-acetylcysteine suppressed oxidative stress and autophagy in RGCs and increased levels of glutathione [308].

Rapamycin, a macrolide antibiotic with antineurodegenerative and neuroprotective properties described in Alzheimer’s and Parkinson’s diseases [309,310], has been found to enhance RGC survival in rodent glaucoma models. It achieves this by inhibiting the production of NO and TNF-α in microglia through the modulation of NF-kB activity and by maintaining Akt phosphorylation to inhibit RGC apoptosis [311].

Geranylgeranylacetone (GGA), a molecule used in the therapy of gastric ulcers and known for its antioxidant features, has been shown to reduce oxidative stress in the retina by triggering thioredoxin and heat shock proteins (Hsp)-72, thereby protecting against apoptosis [312]. In murine models of NTG, GGA has been demonstrated to decrease RGC loss through the upregulation of Hsp-70 and a reduction in caspase-3 and -9 activities [313].

These findings highlight the potential of these antioxidant molecules in providing neuroprotection and preserving retinal health in glaucoma. 

### 4.3. Leber’s Hereditary Optic Neuropathy

#### 4.3.1. General Aspects: Genetics, Clinical Presentation, and Current Therapeutic Options

Leber’s hereditary optic neuropathy (LHON) is a relatively rare disease compared to glaucoma, but it is considered the most frequent mitochondrial DNA (mtDNA) disorder [314]. It follows a maternal inheritance pattern and primarily affects young males, typically presenting between the ages of 15 and 35 [32]. The most prevalent mutations associated with LHON are m.3460G>A, m.11778G>A, and m.14484T>C, which account for approximately 90–95% of all cases [13]. These mutations affect the protein subunits of complex I in the mitochondrial respiratory chain [315]. Among these mutations, m.11778G>A is most frequently observed in Northern Europe, Australia, and Japan [8,12,316,317], while m.14484T>C is more common among the French-Canadian population [318,319,320]. Studies examining the complete penetrance of LHON have reported varying prevalence estimates in Europe, ranging from 1 in 30,000 to 1 in 54,000, approximately 0.00002% of the population [8,9,10,11,12]. A comprehensive Australian review, which accounted for incomplete disease penetrance in variant carriers and used a well-characterized population-based control cohort to minimize sampling bias, estimated a prevalence of 1 in 800 or approximately 0.00125% [13]. Similarly, another study using a similar methodology concluded that LHON prevalence in the general population exceeds 1 in 1000 [14]. These findings suggest that most LHON mutations remain silent until unknown triggers precipitate their conversion from asymptomatic mutation carriers to symptomatic individuals [321]. Indeed, only around 50% of males and 10% of females harboring a pathogenic mtDNA mutation effectively develop the disease [315]. LHON also exhibits a gender bias, with males being more likely to be affected than females [315]. This gender predisposition has been associated with X-linked susceptibility loci [322,323]. Other studies have explored potential factors that may explain the incomplete penetrance and have found that smoking and excessive alcohol consumption are more common in individuals with LHON compared to asymptomatic carriers [324]. Additionally, it has been suggested that vitamin B12 (cobalamin) deficiency could accelerate the symptomatology in LHON carriers [325] and that regular screening for vitamin B12 levels may be considered in LHON carriers and patients [314].

Clinically, LHON is highly disabling and leads to subacute bilateral irreversible vision loss [315]. A recent study examining the quality of life in 17 LHON patients from different countries (Germany, UK, France, and the US) described several daily life challenges in, for example, physical capabilities, interpersonal relationships, work and recreational activities, that significantly impact their well-being [326]. A cost-of-illness study focusing on inherited retinal disorders (IRD) estimated that in 2019, the overall costs associated with LHON amounted to USD 84–200 million and CAD 10–42 million in the US and Canada, respectively [327]. Moreover, healthcare costs, including therapy expenses, represented only 7% and 2% of the total costs in the US and Canada, respectively [327].

It is widely acknowledged that the treatment of LHON should ideally begin within one year from the onset of visual loss [321,328]. In relation to disease progression, it is divided into three stages starting from the onset [329]:Subacute phase (<6 months): During this phase, patients commonly experience blurred vision and impaired color perception without pain, and their pupillary reflex remains unaffected [315]. Approximately 75% of cases initially experience visual loss in one eye, with the contralateral eye becoming affected within a few weeks [315,330]. Fundoscopy may reveal axonal loss in the papillomacular bundle and circumpapillary telangiectasias. Perimetry often demonstrates typical centrocecal or central scotomas [315,330,331]. Optical coherence tomography (OCT) may show swelling of the peripapillary RNFL. Magnetic resonance imaging (MRI) is commonly performed for differential diagnosis [330];Dynamic phase (6–12 months): During this stage, fundoscopic signs such as telangiectasias and RNFL edema gradually regress [330];Chronic phase (>12 months): In the chronic phase, there is a further decline in visual acuity and visual field loss. Fundoscopic examinations may reveal optic nerve head atrophy, while OCT may indicate thinning of the RNFL [330].

An innovative therapeutic approach for LHON has also been developed in the realm of gene therapy. An example of this is lenadogene nolparvovec (Lumevoq©, Gensight Biologics SA, Paris, France), which was administered via intravitreal injections and designed to treat LHON caused by the m.11778G>A variant [332]. This drug utilizes a replication-defective, single-stranded DNA recombinant adeno-associated virus vector of serotype 2. It carries a codon-optimized complementary DNA encoding the human wild-type subunit protein that is affected in the m.11778G>A variant [332]. The viral vector is designed to deliver the therapeutic gene into the targeted cells, aiming to enable them to produce the NADH dehydrogenase 4 (ND4) enzyme, which can cause restoration of the ETC [333,334].

However, it is important to note that this drug has been withdrawn by the EMA as of 20 April 2023, as it was found to be ineffective in improving outcomes for patients with LHON (source: EMA website, https://www.ema.europa.eu/en/medicines/human/withdrawn-applications/lumevoq (accessed on 17 July 2023).

An international consensus has established that idebenone (IDE), a synthetic analogue of coenzyme Q_10_ (COQ_10_), is the first disease-specific antioxidative drug authorized by the European Medicines Agency in 2015. It has been shown to provide benefits when administered during the subacute or dynamic stages of LHON at a dosage of 900 mg/day. However, it is not recommended for use during the chronic phase [328,329,335]. 

Extensive research has been conducted to discover medications that can restore the ETC, which is crucial for ATP synthesis and maintaining a normal redox status [321]. In this context, IDE functions as an electron carrier within the mitochondrial ETC, facilitating electron transfer from complex II to complex III and promoting ATP synthesis [336]. However, it is important to note that Jaber and Polster described IDE as a pro-oxidative molecule that also inhibits complex I [336]. Additionally, Gueven et al. extensively reviewed the pro-oxidative properties of IDE and questioned the actual antioxidative activity of this drug [337]. They compared studies that detected low nanomolar concentrations of IDE in target tissues for only a short period of time with the majority of publications on IDE, which reported the need for micromolar concentrations to achieve an antioxidative effect [338,339,340]. One possible explanation for the apparent contradiction regarding the low bioavailability of IDE is the suggestion of an indirect antioxidant effect through the inhibition of NOX2, which normally produces damaging ROS, such as superoxide [341]. IDE may also activate different signaling pathways that increase the activity of SOD, NAD(P)H quinone oxidoreductase 1 (NQO1), glutathione (GSH), and glutathione peroxidase (GPX), possibly through Nrf2, a transcription factor that regulates the expression of SOD, NQO1, and GPX. Another proposed explanation for the rapid pharmacokinetics of IDE focuses on the possibility that some of its metabolites, such as 6-(9-carboxynonyl)-2,3-dimethoxy-5-methyl-1,4-benzoquinone (QS10), can also provide therapeutic effects [337]. In support of this hypothesis, a previous Italian study concluded that QS10, like IDE, can bypass the defect in complex I and, unlike IDE, can replace the endogenous coenzyme Q_10_ (CoQ_10_), potentially exhibiting even greater activity than IDE in diseases caused by complex I defects or CoQ_10_ deficiency [342]. In Figure 7, direct and indirect effects of idebenone are represented. 

#### 4.3.2. Oxidative Stress in the Pathogenesis of Leber’s Hereditary Optic Neuropathy

Mitochondrial DNA mutations in LHON lead to a defective complex I, resulting in impaired ETC activity [343]. Studies on fibroblast and cybrid mitochondria affected by LHON mutations have shown varying levels of mitochondrial aerobic respiration alterations depending on the specific mutations [344]. Decreased respiration rates have been reported as approximately 20–28% for m.3460G>A, 30–36% for m.11778G>A, and 10–15% for m.14484T>C [344]. Despite the impaired oxidative phosphorylation and reduced ATP production associated with complex I dysfunction, it is proposed that LHON partially compensates for these deficiencies through glycolysis, an alternative energetic pathway observed in human tissues. This compensation is suggested to maintain the total cellular ATP concentration despite the severe decrease in complex I-related ATP synthesis [343,345]. Hence, additional processes need to be considered to explain the clinical manifestation of the disease. Carelli et al. proposed that decreased pH resulting from defective complex I affects redox sites during aerobic respiration, leading to the overproduction of ROS [345]. Studies on cells carrying LHON mutations have reported increased levels of oxidative stress biomarkers, such as 8-OHdG, the reduced activity of antioxidant systems, including glutathione reductase and Mn-SOD, and elevated levels of oxidized glutathione (GSSG) [346,347,348]. Lin et al. demonstrated in a murine model of mtDNA mutations that LHON mutations cause a systemic elevation in ROS production, with chronic elevation in ROS generation observed in synaptosomes, optic nerves, and RGCs, suggesting a more significant damaging role of oxidative stress compared to ATP depletion [349]. Another study in mice aimed at establishing a murine model of LHON found that a mutation in the subunit ND4 of complex I led to mitochondrial structure disassembly, increased ROS levels, ONH swelling, and RGC apoptosis [350]. Additionally, an alternative hypothesis suggests that altered permeability transition pores, possibly due to Ca^2+^ and ROS-mediated depolarization of the mitochondrial membrane, may play a role in LHON. These opened permeability transition pores in cybrid cells carrying LHON mutations could facilitate the release of cytochrome c from mitochondria to the cytosol, activating the apoptotic pathway [351].

Dysfunctional complex I in LHON may lead to sensitization of permeability transition pores primarily due to increased ROS generation and cytosolic Ca^2+^ overload. The reduced ATP supply to the Ca^2+^ pumps in the ER membranes may contribute to the intracellular Ca^2+^ overload by impairing Ca^2+^ uptake from the cytosol to the ER [351]. Consequently, the death of RGCs in LHON is likely influenced by the altered redox status and intracellular Ca^2+^ overload, with an emphasis on the role of ROS in triggering proapoptotic pathways (Figure 8) [349,352,353,354,355,356]. Apoptosis in LHON has been described as mediated by caspases, including the Fas-induced pathway involving caspase-8 [355,356]. The activation of caspase-8 leads to the cleavage of protein Bid (BH3 interacting-domain death agonist), a component of the Bcl-2 family, indirectly causing the release of cytochrome c from mitochondria. Cytochrome c then binds to Apaf-1, along with caspase-9, forming a complex called the “apoptosome.” The apoptosome subsequently activates caspase-3, which is responsible for cellular disassembly and apoptosis. Additionally, caspase-independent pathways have been observed in LHON, involving the release of cytochrome c, AIF (apoptosis-inducing factor), and EndoG (endonuclease G) into the cytosol [357]. These apoptotic processes contribute to the demise of RGCs in LHON. 

### 4.4. Therapeutic Perspectives in Leber’s Hereditary Optic Neuropathy

α-Tocotrienol quinone, also known as EPI-743, is a synthetic molecule classified as a para-benzoquinone. It targets NQO1 and leads to the replenishment of intracellular glutathione stores, thereby increasing antioxidant capabilities [358]. In vitro studies have shown that EPI-743 is over one thousand-fold more effective than IDE in protecting cells from oxidative stress [359]. An investigation into the effect of EPI-743 in LHON demonstrated disease progression arrest and the reversal of vision loss in four out of five patients treated with EPI-743 within 4 months of the onset of visual loss [360]. Another study involving 12 patients with active phase LHON concluded that EPI-743 stabilized or improved visual function in approximately 70% of the treated eyes [361]. However, larger trials in the LHON population are needed to further validate these findings. Currently, EPI-743 is in phase II of various clinical trials (NCT01370447; NCT04378075; and NCT02352896) aimed at evaluating its effectiveness in primary mitochondrial disorders [362].

Elamipretide, also known as MTP-131, Bendavia, or SS-31, is a relatively new mitochondria-targeting peptide that has been shown to protect RGCs from oxidative stress-induced apoptosis [363]. Other interesting molecules that modulate mitochondrial redox status and enhance mitochondrial biogenesis include sonlicromanol (also known as KH176) and KL1333 [362]. KL1333 is an NAD^+^ modulator that improves the NAD^+^/NADH ratio and has been demonstrated to decrease lactate and ROS levels while increasing ATP synthesis in fibroblasts from patients with MELAS (mitochondrial encephalopathy, lactic acidosis, and stroke-like episodes) [364].

These emerging therapeutic options provide potential avenues for the treatment of LHON by targeting oxidative stress and mitochondrial dysfunction. 

### 4.5. Anterior Ischemic Optic Neuropathy

#### 4.5.1. General Aspects: Prevalence, Clinical Presentation, and Current Therapies

Among optic nerve disorders, anterior ischemic optic neuropathies stand out as one of the most complex and significant conditions. These conditions are classified into anterior and posterior optic neuropathies (AION and PION), distinguished by the presence or absence of disc edema, respectively [365]. They are further categorized into arteritic and nonarteritic diseases based on the presence or absence of vasculitis, which results in reduced perfusion to the optic nerve head [366]. Nonarteritic anterior optic neuropathy (NA-AION) is the most common acute optic nerve disease in patients older than 50 years [367]. Clinical presentations include sudden acute unilateral vision loss without pain. However, effective treatments for these conditions remain challenging [16,367,368]. Giant cell arteritis (GCA) is the most common subtype of arteritic anterior ischemic optic neuropathy (A-AION). High intravenous corticosteroid doses serve as the first-line therapy for GCA [369]. It is crucial to initiate rapid treatment once GCA is suspected [370]. A systematic review predicts that by 2050, over 3 million individuals in Europe, North America, and Oceania will suffer from giant cell arteritis primarily due to aging [371]. Additionally, the estimated economic burden resulting from visual disability caused by GCA in the United States is expected to exceed USD 76 billion by 2050 [371]. Clinically, GCA manifests in individuals over 50 years old, presenting classic symptoms such as severe headaches, jaw claudication, cutaneous allodynia, and, in the majority of patients, additional symptoms like fever, weight loss, depression, and night sweats [370]. Despite the administration of high steroid doses, including intravenous methylprednisolone 1000 mg/day for 3 days, followed by a maintenance dose of 1 mg/kg of prednisone [372], the chances of visual recovery in GCA-related visual loss remain very low [373]. Furthermore, the long-term use of steroids is associated with various side effects and rebound syndrome, while GCA is prone to relapse [374]. Consequently, there is a significant research need for new therapeutic strategies, and immunomodulating therapies have emerged as promising avenues in the treatment of GCA. One example of an immunomodulating therapy is tocilizumab (TCZ), an interleukin-6 (IL-6) receptor antibody. TCZ has already received approval from the National Institute for Health and Care Excellence and NHS England for the treatment of refractory GCA [375]. However, the use of TCZ is restricted due to its known side effects, including alterations in liver enzymes and cholesterol levels, gastrointestinal perforation, infections, headaches, and arterial hypertension [376,377]. In the field of immunomodulation, several new immunoglobulins are currently undergoing clinical trials for the treatment of GCA [378].

Nonarteritic anterior ischemic optic neuropathies (NA-AION) occur as a result of events that lead to hypoperfusion of the optic nerve head. A recent comprehensive meta-analysis revealed several risk factors for NA-AION, including male gender, hypertension, hyperlipidemia, diabetes mellitus, coronary heart disease, sleep apnea, factor V Leiden heterozygosity, and a history of cardiovascular medication use [379]. Pathogenetically, Hayreh SS. described that NA-AION involves hypoperfusion of the optic nerve head, leading to hypoxia of the retinal ganglion cell axons. This, in turn, results in the stasis of the axoplasmic flow and the generation of swollen axons [366]. Consequently, optic disc edema occurs, causing compression of the capillaries supplying the optic nerve head, creating a vicious cycle. The primary trigger for the initial hypoperfusion of the optic nerve head is typically a transient drop in blood pressure, often occurring during sleep, such as nocturnal arterial hypotension or hypotension after sleeping during the day. Severe occlusion of the internal carotid or ophthalmic artery rarely causes ocular ischemia in NA-AION cases [366]. While embolic lesions can be an occasional cause of NA-AION, they result in more extensive and permanent damage to the optic nerve head compared to the hypotensive type [366]. The clinical onset of NA-AION is sudden and typically affects a single eye, resulting in vision loss without pain, often discovered by patients upon waking in the morning [380]. Fundoscopy and perimetry are essential diagnostic tools for NA-AION. The Goldmann perimeter, in particular, helps detect classic visual field defects in NA-AION. A study on 312 consecutive eyes reported central scotomas in nearly half of the cases, with the absolute inferior nasal defect being the most commonly detected [381].

Therapeutic options for NA-AION remain a challenging clinical issue. The effectiveness of steroids, whether administered orally or intravenously, is a subject of intensive debate [382]. A large nonrandomized cohort study in the United States involving 696 eyes, spanning from 1973 to 2000, suggested that systemic steroid treatment with oral prednisone (80 mg per day) during the acute phase may increase the probability of visual acuity and visual field improvement after six months [383]. However, due to the nonrandomized nature of this study and potential biases, the findings should be interpreted with caution [384]. Katz and Trobe extensively reviewed possible treatment strategies for NA-AION in both conservative and surgical fields and concluded that optic nerve fenestration surgery is ineffective and potentially harmful, while the efficacy of steroids remains uncertain [385]. In an Indian randomized controlled trial involving 38 patients with NA-AION, Saxena et al. found that oral prednisolone at 80 mg per day reduced the duration of disc edema and improved the electrophysiological parameters of the optic nerve but did not result in a visual acuity benefit after six months [386]. One potential clinical approach is the use of α_2_-adrenergic agonists, such as brimonidine, in the acute phase of AION through topical application. A study investigating the neuroprotective effect of brimonidine on anterior optic neuropathy in rodents (rAION) demonstrated a decrease in RGC loss in mice treated with brimonidine in the acute phase, possibly through a reduction in VEGF-A and HIF-1α expression [387]. These findings align with previous studies that also demonstrated the effectiveness of brimonidine in the acute phase of rAION [388,389]. In summary, the significant lack of effective treatments for NA-AION and its relatively high prevalence underscore the urgent need for research in novel therapeutic approaches. Exploring the antioxidative branch of treatment holds promising potential to make significant contributions in this regard. By focusing on antioxidant strategies, new therapeutic avenues can be explored to address the challenges posed by NA-AION.

#### 4.5.2. Oxidative Stress in the Pathogenesis of Anterior Ischemic Optic Neuropathy

GCA is an autoimmune disorder characterized by granulomatous infiltration involving T-cells and activated macrophages, including multinucleated giant cells, primarily observed in the vascular wall of the aorta and its main branches [370,390,391,392]. Vascular aging is considered one of the critical risk factors for GCA, as oxidative stress is believed to play a significant role in its pathogenesis [370]. This is likely attributed to age-associated mitochondrial dysfunction, leading to an increased ROS generation in endothelial cells and vascular smooth muscle cells [391,393,394]. A study on GCA patients has demonstrated the presence of neutrophils in the vascular lumen and adventitia tissues surrounding the external lamina elastica of temporal arteries, producing high levels of extracellular ROS [395]. Additionally, systemic oxidative stress parameters, such as TAC and MDA levels, along with intracellular leukocyte ROS levels, were found to be elevated in GCA patients [396], and the findings support the hypothesis that reactive species contribute to the pathophysiology of GCA by inducing vascular stress in large and medium-sized vessels.

In the pathogenesis of NA-AION, damage may arise as a consequence of hypoxia in the optic nerve head. Cellular hypoxia resulting from glucose and oxygen depletion, followed by reoxygenation during the ischemia/reperfusion (I/R) process, can lead to an overproduction of reactive species [397,398,399,400]. It has been proposed that systemic corticosteroids are effective in the acute phase of NA-AION, as they suppress NF-kB activation during inflammation [383]. NF-kB is a transcription factor that can be triggered by ROS-related cascades, as described in the pathogenesis of glaucoma [40,208,401]. ROS have been shown to cause immune-mediated neuronal injuries, disruption of the blood–optic nerve barrier, apoptosis, and autophagy through damage to DNA, proteins, and lipids [402,403]. Studies have revealed that I/R injuries in ischemic neurons lead to somatic autophagy of axonal mitochondria, resulting in increased retrograde movement and decreased anterograde movement of mitochondria, ultimately reducing the functional mitochondria available for ATP synthesis [403]. Our laboratory’s study on short-term ischemia in porcine models demonstrated hypoxia-related alterations characterized by the upregulation of HIF-1α, VEGF-A, NOX2, iNOS, and high levels of ROS [143]. Supporting these findings, a study on NOX2-deficient murine models (NOX2^−/−^) showed that these mice displayed neuroprotection in retinal I/R injury scenarios [228].

Interestingly, a Turkish study investigating plasma samples from 18 newly diagnosed NA-AION patients found no significant differences in systemic oxidative stress parameters, including TOS and TAS, when compared to healthy controls [402]. However, genetic investigations have revealed an association between NA-AION and a loss-of-function deletion in the gene GSTM1, which encodes one of the three isoforms of the antioxidative enzyme glutathione-S-transferase (GST) [404,405]. Additionally, another study reported a higher frequency of mitochondrial DNA mutations in NA-AION patients compared to controls, suggesting that mitochondrial dysfunction may be a risk factor for NA-AION [406].

Based on the current literature and the pivotal role of hypoxia in NA-AION, some studies have provided evidence that conditions leading to increased ROS generation may contribute to the risk of developing NA-AION. These findings highlight the potential involvement of ROS in the pathogenesis of this disease.

### 4.6. Therapeutic Perspectives in Anterior Ischemic Optic Neuropathy

#### 4.6.1. Giant Cell Arteritis

Nuclear sirtuins, such as the before-mentioned SIRT1, are proteins that potentially have an antioxidative impact on GCA. These enzymes inhibit inflammation and oxidative stress by transcriptionally repressing various inflammation-related genes [407,408]. A recent study observed SIRT1 downregulation and subsequent ROS generation in individuals with GCA, suggesting a potential therapeutic effect of SIRT1 activators [396]. While immunomodulating molecules have been extensively investigated due to the autoimmune nature of GCA, there is growing scientific interest in exploring new antioxidant molecules such as nuclear sirtuins.

#### 4.6.2. Nonarteritic Anterior Ischemic Optic Neuropathy 

An investigation using rAION mice and intravitreal injections of the Rho-kinase inhibitor, E212, immediately after optic nerve infarction resulted in increased SOD activity, decreased ROS levels, reduced oxidative stress, and preservation of the blood–retinal barrier [409]. Our laboratory conducted a study on mice with I/R retinal injuries and found that betulinic acid, a natural compound found in plane bark, leaves, and fruit peel, preserved vascular function, attenuated ROS formation, and upregulated SOD and HO-1 [131]. Oroxylin A, a bioactive flavonoid extracted from Scutellariae baicalensis Georgi, showed potential protective effects in rAION by activating Nrf2, increasing HO-1 and NQO1 activity, promoting RGC survival, maintaining RNFL thickness, and exhibiting anti-inflammatory effects by reducing IL-6 and increasing TGF-β levels [410]. N-butylidenephthalide, another molecule, demonstrated a neuroprotective role in rAION by inhibiting the NF-κB signaling pathway [411]. Vitamin B3 (niacin) was found to provide neuroprotection from oxidative stress in rAION by activating Nrf2, increasing SOD expression, inhibiting mitochondrial apoptosis, and reducing the inflammatory response through NF-κB inhibition [368].

Geranylgeranylacetone (GGA), previously mentioned as a promising molecule for glaucoma therapies, was also tested in murine models of I/R retinal injury and demonstrated a decrease in RGC death by inhibiting the p38 MAPK apoptotic pathway [412]. In a more recent study, GGA was shown to induce an increase in Hsp70 levels, inhibiting glial activation, autophagy, and apoptosis [413]. Astaxanthin, previously identified as a potential antiglaucomatous drug, possesses antioxidant and antiapoptotic properties. Lin et al. found that astaxanthin preserved visual function, increased RGC survival, and inhibited apoptosis by blocking the Akt pathway in rAION [414]. Resveratrol, previously investigated for its potential in preventing RGC loss in glaucoma, has also been studied in I/R injuries. It was found to attenuate glial activation and RGC death by suppressing HIF-1α-VEGF-A upregulation and activating the downregulation of the PI3K/Akt pathway [415]. Resveratrol also protects RGCs from ischemia-related injuries by increasing Opa1 expression [416]. Our recent research demonstrated that resveratrol prevents vascular dysfunction and RGC death in I/R-induced murine models, possibly through the inhibition of the I/R-related upregulation of NOX2 [98]. These findings highlight potential therapeutic approaches for AION by targeting oxidative stress, inflammation, and mitochondrial dysfunction.

### 4.7. Optic Neuritis

#### 4.7.1. General Aspects

Optic neuritis (ON) is an idiopathic inflammatory demyelinating disease of the optic nerve [417,418], which can be associated with the neuromyelitis optica [419] or can also appear as initial manifestation of multiple sclerosis, a chronic demyelinating disorder of the central nervous system [420,421]. In addition, reported risk factors for ON are granulomatous diseases, infections, and autoimmune pathologies [422]. In the pathogenesis of ON are involved activated T-cells, which releasing an abnormal volume of inflammatory cytokines, elicit demyelination and disruption of the blood–optic nerve barrier and lead to the loss of retinal ganglion cells and ONH atrophy [44,419,423,424,425]. ON is described as the major cause of acute optic nerve disorder in young patients [419], being commonly observed between 20 and 45 years and predominantly in women, with a ratio of 3:1 [17,420,426]. Its clinical presentation can occur through single or multiple episodes and usually consists of a sudden and unilateral loss of visual acuity, afferent pupillary impairment, orbital pain by eye movements and altered color perception, in the presence or absence of disc edema [427]. Diagnostically, fundoscopy and OCT help to detect ONH-anomalies and RNFL thinning, respectively. A study estimated that OCT can reveal thinning of the RNFL, with an average reduction of 33% compared to controls and an average reduction of 27% between impaired and unimpaired eyes of the same patient [428]. Furthermore, it has been reported that RNFL thinning is visible in 85% of all patients suffering from ON within 3–6 months after the acute phase [429]. In addition, perimetry can evidence visual field loss limited to the nerve fiber bundle region, with paracentral, partial arcuate, or arcuate defects [430]. A cranial MRI scan is performed for differential diagnosis and to evaluate the association with multiple sclerosis [420]. Current standard therapies of ON include high intravenous steroid doses (1000 mg IV methylprednisolone) during acute phases, which were reported to facilitate visual recovery, whereas oral prednisone alone is not recommended, as it failed to show comparable improvements [431]. However, a systematic review examining six clinical trials globally, including 750 patients affected by ON, did not evidence beneficial effects neither through intravenous nor via oral steroid therapy, compared with the placebo groups, in terms of the visual field, visual acuity, and contrast sensitivity outcomes [432]. Alternatively, the use of intravenous immunoglobulins in acute ON, which may be considered in steroid-refractory cases [433], failed anyway to provide benefits for generalized practice [434,435].

#### 4.7.2. Oxidative Stress in the Pathogenesis of Optic Neuritis

Various investigations succeeded to provide evidence on an altered redox status in the context of ON, using experimental autoimmune encephalomyelitis (EAE) models [411,436,437,438] and an animal model originally established for multiple sclerosis, often employed in experiments aimed to study optic neuritis [427]. Moreover, studies on patients affected by ON showed significant ROS-related anomalies, such as an elevated fraction of oxidized thiol [439] or reduced bilirubin serum levels [440], indicators of high ROS generation and low antioxidant status, respectively. As mentioned before, similar to NA-AION and LHON, ON loss-of-function deletions in the gene coding for the antioxidative enzyme glutathione-S-transferase were also detected [405], therefore emphasizing a decreased antioxidant activity as a possible risk factor. As we previously highlighted in glaucoma, an elevated ROS generation and active inflammatory grade are deeply interconnected and tend to trigger and intensify each other in a process where NF-kB and cytokines are fundamental pathomechanistic players [441]. In relation to ON, increased ROS are responsible for myelin phagocytosis and stimulate changes in the permeability of the blood–brain barrier, favoring migration and infiltration of active T-cells, which, in turn, reflects in abnormal inflammation and augmented ROS production [439]. Oxidative damage may possibly be an initiator for demyelination and neurodegeneration [439], which finally drives to RGC loss [417].

More large studies designed to interpret the multiple aspects in the pathogenesis of ON, including the role of oxidative stress, may deliver new knowledge for the design of more effective pharmacological drugs to treat this disease.

### 4.8. Therapeutic Perspectives in Optic Neuritis

While a prompt administration of anti-inflammatory drugs during the acute phase of optic neuritis (ON) may help preserve vision [442], the use of steroids alone has not been shown to protect the optic nerve from inflammatory demyelination or prevent degeneration of retinal ganglion cells (RGCs) [417,443]. Therefore, antioxidant approaches have been explored to support nerve fiber integrity by counteracting inflammation and demyelination, thus preventing RGC death [417,444]. In preclinical studies on ON models, several major antioxidant targets have been investigated, including the activation of Nrf2 and the suppression of ROS-related proapoptotic signaling, such as blocking ASK-1 and the pJNK/p38 pathway.

Dimethyl fumarate, a Nrf2 activator that has been licensed for the treatment of relapsing-remitting multiple sclerosis in the US since 2013, has shown promising results in experimental autoimmune encephalomyelitis (EAE) models, reducing the severity and relapses of optic neuritis and preserving RGCs from cell death [444]. Gypenosides, an extract from Gynostemma pentaphyllum, have demonstrated beneficial antioxidant effects on the retina [445]. In a study on murine models of optic neuritis, gypenosides led to a decrease in iNOS and COX2 expression while activating Nrf2, resulting in the upregulation of HO and GPX, along with free radical scavenging and anti-inflammatory activities [446]. α-Lipoic acid, which potentially activates Nrf2 [268], has been investigated in EAE models, showing increased RGC survival and anti-inflammatory effects [447]. However, a dedicated trial on 31 patients with ON (NCT01294176) reported good tolerability after oral supplementation (600 mg twice a day) of α-lipoic acid [448] but did not show significant benefits in retinal nerve fiber layer thickness after 24 weeks compared to the placebo group [449]. Consequently, the trial ended in phase I due to insufficient recruitment [450]. As previously mentioned, SIRT1, known to activate Nrf2, has also been studied [417]. In an EAE study, an adeno-associated virus vector was utilized as a delivery vehicle to enhance SIRT1 expression, demonstrating that the selective upregulation of SIRT1 promotes RGC survival and preserves axons from demyelination through intravitreal injection [417]. Another investigation explored the use of matrine, an extract from the herb Radix Sophorae Flavescentis, in EAE models. Matrine induced an overexpression of SIRT1, the subsequent upregulation of Nrf2, and collectively resulted in mitochondrial biosynthesis, reduced ROS formation, and the suppression of inflammation and demyelination [451].

The inhibition of ASK-1 also showed encouraging preclinical results in mitigating oxidative-stress-induced RGC apoptosis. The ASK-1 inhibitor MSC2032964A has been demonstrated to alleviate neuroinflammation and diminish optic nerve demyelination in EAE murine models [452]. Spermidine, previously mentioned as a potential antioxidant for glaucoma, also suppresses the ASK-1/p38/pJNK pathway [263] and has been reported to reduce RGC apoptosis in EAE mice [453].

A recent publication investigated the effects of edaravone, previously mentioned, in cultures and mice with neuromyelitis optica spectrum disorder and found that it promoted remyelination through activation of the mTOR complex I (mTORC1) signaling pathway [454]. Currently, there is an ongoing trial (NCT05540262) that aims to assess the impact of edaravone in patients with optic neuritis who are positive for aquaporin-4 antibodies. The trial is expected to be completed by the end of 2024.

## 5. Other Rare Optic Neuropathies

### 5.1. Traumatic Optic Neuropathies

#### 5.1.1. General Characteristics

Craniofacial traumas can directly or indirectly insult the optic nerve, leading to so-called traumatic optic neuropathies (TONs) [455,456]. Direct traumas are caused by injuries to the optic nerve through the intracranial fragmentation of bones or also through contusion, causing anatomical disruption [457]. Indirect traumas occur when compression and disruption of pial vessels cause a reduction in the vascular perfusion in the optic nerve [458,459]. Moreover, a deformation of the skull at the optic canal, and more specifically at its intracranial end, was reported to also be responsible for optic nerve injuries in indirect traumas [460]. The most reported causes of TON are vehicle accidents, bicycle accidents, falls, assaults, and sport injuries [24]. Indirect TON is more common than direct forms and occurs in 0.5–5% of all closed head injuries and 2.5% of all midfacial fractures [22,461,462]. In terms of diagnosis, a brain MRI scan is typically conducted [456]. Therapeutically, reported options for the treatment of traumatic optic neuropathy (TON) include high-dose corticosteroids or surgical decompressions [456]. However, both of these treatment approaches are accompanied by controversies due to the pharmacological side effects of steroids or the potential complications associated with surgery [456,463]. Consequently, exploring neuroprotective strategies to prevent optic nerve damage represents a promising curative option with substantial potential for TON [456].

An emerging and intriguing area of research for traumatic and inflammatory optic nerve disorders is neuromodulation based on biomedical ultrasound stimulation. This field is continuously evolving, not only for the retina but also for other ocular structures, such as the cornea and ciliary body [464]. Moreover, it extends beyond ophthalmology, finding applications in fields like cardiology and the peripheral nervous system [465,466,467,468,469]. Retinal ultrasound stimulation is a promising therapeutic approach for TON [470]. Some studies reported that curative strategies based on ultrasound may prevent oxidative-stress-induced damage [471,472], even on retinal pigmental epithelial cells in vitro [473]. In cases of complete RGC degeneration resulting from severe optic nerve trauma, ultrasound neuromodulation may offer a potential noninvasive approach to restore vision. High-frequency ultrasound can biochemically initiate localized neuronal responses, triggering electrically evoked cortical potentials along the visual pathway [470,474]. Implantable treatments, including a retinal stimulating piezo-array, are also being explored as wireless retinal prostheses capable of eliciting visual percepts in cases of irreversible visual loss [475]. A study conducted on rabbits demonstrated that an intraneural electrode array positioned in the intracranial segment of the optic nerve can generate selective activation patterns in the visual cortex, sparking interest in the potential use of optic nerve prostheses for total retinal detachment cases [474]. Another intriguing possibility is direct stimulation of the visual cortex bypassing the visual pathway, which was recently explored using hybrid noninvasive ultrasound techniques [476].

#### 5.1.2. Pathogenesis of Traumatic Optic Neuropathies: Role of Oxidative Stress and Ca^2+^

In murine models, traumatic optic neuropathy (TON) has been shown to be associated with RGC loss, axonal degeneration, and visual impairment [477]. The pathomechanisms potentially involved in RGC death include the disruption of the blood–brain barrier, leading to the infiltration of activated immune cells, such as macrophages and neutrophils. These immune cells produce a significant amount of reactive oxygen species (ROS) and cytokines, further activating microglia and causing damage to mitochondria and the endoplasmic reticulum [478,479,480]. Numerous studies on murine and rodent models of TON have demonstrated an overproduction of ROS [481,482,483], as well as a decrease in the antioxidant activity of SOD [470] and CAT [484,485]. Additionally, mitochondrial anomalies [486] and disturbances in Ca^2+^ fluxes, which can occur concomitantly with ROS formation, have been described [485]. Notably, the progression of damage in TON has been associated with intracellular Ca^2+^ influx, observed after stretching axonal myelinated fibers of the optic nerve, resulting in the dissolution of the axonal cytoskeleton [487]. An insightful review discussed the interplay between endoplasmic reticulum (ER) stress and oxidative stress in the pathophysiology of TON, highlighting the involvement of these two pathogenic factors in optic nerve trauma-induced RGC loss [488]. In this context, a study by Hu et al. on optic nerve crush models revealed an upregulation of the ER stress mediator protein CHOP as a consequence of optic nerve injury, further demonstrating that its deletion can prevent RGC death [489]. Consistent with these findings, other investigations on models of traumatic brain injury have reported the overexpression of CHOP and induction of ER stress, contributing to neurodegenerative processes following neurotrauma [490,491,492]. In summary, in line with existing literature, after an optic nerve insult, disruption of the blood–optic nerve barrier may allow the infiltration of activated immune cells, resulting in the generation of high concentrations of ROS and proinflammatory mediators. These events ultimately lead to axonal degeneration, myelin damage, and culminate in mitochondrial and ER stress, further exacerbating inflammation and triggering apoptotic RGC death.

#### 5.1.3. Potential Antioxidants for Traumatic Optic Neuropathies

Numerous naturally occurring compounds have been tested for their antioxidant properties in models of traumatic optic neuropathy (TON). For example, the ethanol extract of Echium amoenum L. demonstrated antioxidant and anti-inflammatory effects in optic nerve crush models. It was shown to inhibit glutamate-induced ROS formation, reduce NF-kB activity, and decrease microglial activation and optic nerve damage [493]. Similarly, the ethanol extract of Lithospermum erythrorhizon in optic nerve crush models was found to decrease ROS generation and lower the levels of proapoptotic proteins, such as caspase-3 [494]. In experimental neurotrauma murine models, a high vitamin E diet exhibited neuroprotective properties by preventing RGC death, reducing ROS levels, and suppressing inflammasome activation [484]. Neuroglobin, a mammalian heme protein with antioxidant features, was reported to enhance RGC survival and promote optic axon regeneration in TON murine models [495,496,497].

Moreover, synthetic compounds have also been investigated in TON models. Carvedilol, a nonselective β-adrenoreceptor blocker, was found to inhibit iNOS expression, ASK-1, and the p38/MAPK pathways, subsequently reducing RGC apoptosis [498]. Polydopamine is a polymer synthetically obtained through oxidation of the neurotransmitter dopamine [499] and a major pigment of the naturally occurring eumelanin [500]. Polydopamine-based nanoreactors were demonstrated to scavenge ROS in RGCs after optic nerve injury in mice, as well as to mitigate microglial activation and to suppress ROS-related RGC apoptosis [501]. Galantamine is an FDA-licensed compound used in the treatment of Alzheimer’s disease, which primarily acts as an acetylcholinesterase inhibitor but also has antioxidant [502,503] and anti-inflammatory features [504]. This molecule was reported to protect against visual function deficiency, attenuate oxidative stress and inflammation, and inhibit axonal degeneration in TON models [505]. Finally, the previously reported resveratrol, which can activate SIRT1 and subsequently, Nrf2 [256,257], also appears to possess curative effects in TON models. In fact, this compound showed protective effects on RGCs in optic nerve crush models, attenuating oxidative stress [506]. In line with these findings, a recent investigation revealed that the regulation of the SIRT1/mTORC1 axis in the microglia has the potential to mitigate optic-nerve-crush-induced RGC death [507].

Taken together, many publications have gathered evidence on promising preclinical results in animal models of optic nerve trauma. However, an important factor to be considered for the practical use of antioxidants after optic nerve trauma in humans is the prompt administration of the drug after diagnosis [478]. A delay in this context may result in unsuccessful effects of the medication due to the early activation of intracellular downstream transductions [478]. Thus, a combination of antioxidants and other medications with possibly different targets may enhance the probability of effectiveness in contrasting damage after optic nerve injuries [478]. Further investigations are warranted to test the efficacy and safety of these molecules.

### 5.2. Compressive and Stretch Optic Neuropathies

#### 5.2.1. General Features

Graves orbitopathy (GO) can cause dysthyroid optic neuropathy (DON), which is due to optic nerve compression (found in over 90% of cases) through the extension of extraocular muscles, orbital fat expansion and interstitial edema [508], or alternatively, optic nerve stretching without compression (described in less than 10% of cases) [509]. DON is recognized as the most common form of compressive optic neuropathy, and its main reported risk factors comprise advanced age, male gender, smoking, and diabetes mellitus [509]. This disorder clinically manifests with central vision defects, altered color perception, relative afferent pupillary defect, and mostly and typically through protrusio bulbi, also known as exophthalmos, due to widening of the extraocular muscles, which leads to restricted ocular motility [509]. The diagnostic features include (1) fundoscopy, which can evidence compression-induced disc edema; (2) perimetry, which help detect central scotomas; and (3) OCT, which can show RNFL thinning. Further, cranial MRI or CT scans may help to differentiate compressive from stretch conditions [510]. The gold standard therapy in DON caused by compression is a high intravenous dose of methylprednisolone (0.5–1 g for 3 days), eventually followed by surgical decompression in the case of insufficient corticosteroid response [511]. In the case of DON caused by stretching, the surgical option becomes the first-line treatment [509]. Other reported therapeutic options are biological drugs, such as teprotumumab and tocilizumab, and orbital radiotherapy [511]. Although first-line therapies are largely assumed to be effective to control symptoms and to prevent vision loss in DON, deterioration of the optic nerve function, relapses of DON, and corticosteroid side effects or contraindications, induce the search for new curative options [511].

#### 5.2.2. Oxidative Stress in the Pathogenesis of Dysthyroid Optic Neuropathy

DON is a consequence of GO, an autoimmune inflammatory disorder, in which through the stimulation of autoantibodies and interactions with activated T-cells, orbital fibroblasts are triggered and induce diverse intracellular transductions, such as the MAPK pathway with downstream NF-kB activation, as well as the PI3K/Akt axis, collectively inducing the abnormal generation of proinflammatory cytokines, such as IL-1α, IL-1β, IL-6, IL-8, macrophage chemoattractant protein-1 (MCP-1), and transforming growth factor (TGF)-β [508,512]. Moreover, orbital fibroblasts differentiate into adipocytes and myofibroblasts, further producing glycosaminoglycan, a component of connective tissues, responsible, through its deposition, for muscle enlargement, together with hyaluronic acid [513]. Investigations have also demonstrated that indicators of oxidative damage are increased in blood and urine, as well as in the orbital fibroblasts of patients with GO. In the blood of patients affected by GO, elevated levels of H_2_O_2_ were demonstrated as well as of lipid hydroperoxides (ROOHs) and the further enhanced activities of SOD and CAT, while GPX activity was reduced [514]. Biomarkers of oxidative stress, such as 8-OHdG and MDA, as well as intracellular ROS were elevated in the orbital fibroblasts of patients with GO compared to healthy controls [515]. High concentrations of MDA and 8-OHdG were also found in the tears of patients with GO, particularly in the active phases of the disease [516]. Another study reported that positive urinary 8-OHdG correlates with disease activity and that smoking as risk factor has a higher influence on the elevation of 8-OHdG [517]. In this regard, an in vitro investigation on the orbital fibroblasts of patients affected by GO, exposed to cigarette smoke extracts, showed that fibroblasts react to cigarette smoke components through the aberrant inducement of oxidative stress, as well as through the elevation of TGF-β and IL-1 [518]. These studies globally evidenced that oxidative stress is present in GO and possibly indicate that ROS have a role in the pathophysiology of this disorder. In this context, hypotheses on the impact of ROS in the pathogenesis of GO were formulated and proven in some studies. For example, O_2_^•−^ was suggested to trigger a retro-ocular fibroblast proliferation in patients with GO [519]. In addition, H_2_O_2_ was shown to stimulate the production of TGF-β and IL-1 in GO orbital fibroblasts [520].

Altogether, from the existing literature, we can extrapolate that ROS, IL-1, and TGF-β play critical roles in the stimulation and progression of inflammation, in the fibroblastic differentiation, and in the deposition of connective tissue components in GO, which collectively lead to remodeling events that cause compression of the optic nerve in DON [521].

#### 5.2.3. Antioxidant Candidates in Dysthyroid Optic Neuropathy

Some molecules with antioxidant characteristics showed encouraging results in dedicated investigations for mild stages of GO [508]. A relevant example is represented by selenium, a natural compound with antioxidant and anti-inflammatory properties [522], present in nuts, shrimps, eggs, meat, and cereals, among others [523]. This molecule was assessed in the orbital fibroblasts of patients with GO and was shown to inhibit fibroblastic proliferation and release proinflammatory cytokines, such as TNF-α, and the ROS-induced production of hyaluronic acid [524]. In line with these findings, another in vitro investigation on orbital fibroblasts affected by GO confirmed the suppressing action of selenium against ROS formation, hyaluronan production, and the release of inflammatory mediators [525]. A dedicated trial on 159 patients with mild GO tested the potential curative effects of selenium supplementations and proved an improvement in symptoms as well as quality of life, together with a slowed-down disease progression after selenium-based therapy, ultimately recommending a six-month treatment through selenium supplementation in patients with mild GO of short duration [526,527]. However, the administration of selenium should be considered in relation to selenium intake from the diet, which, in turn, varies in accordance with different geographical locations [523]. Hence, the detection of selenium concentrations in serum should occur preliminary to prevent a high selenium intake, which may also produce side effects [521,528].

Quercetin, an antioxidant flavonoid found, for example, in fruits and vegetables and used in traditional Chinese medicine [529], was reported to decrease ROS formation as well as to inhibit adipogenesis in primary cultured orbital fibroblasts obtained from GO patients exposed to cigarette smoke extracts [530]. In addition to the demonstrated antioxidative effect, quercetin was also described as a significant anti-inflammatory molecule in orbital fibroblasts affected by GO, suppressing IL-1-related inflammation as well as hyaluronan formation and adipogenetic processes [531].

Other natural compounds which also showed an effective antioxidant activity in orbital fibroblasts from patients with GO in preclinical studies are ascorbic acid in combination with N-acetyl-L-cysteine and melatonin [532] and β-carotene [533]. Furthermore, pentoxifylline was tested in preclinical [534] and clinical investigations [535,536], showing the promising effects contrasting the glycosaminoglycan production from orbital fibroblasts in both. However, the same trial which the confirmed curative effects of selenium in 159 patients with GO, also tested pentoxifylline and did not conclude an equivalent effectiveness for this class of molecules in GO [526].

The before-mentioned resveratrol was also investigated in fibroblasts with GO, showing a decrease in ROS levels and suppressing adipogenesis in vitro [537].

Additionally, synthetic existing compounds were tested as potential antioxidants in GO. In this regard, allopurinol in combination with nicotinamide showed promising results in vitro, reducing ROS formation in fibroblasts [538], as well as in clinical trials, positive impacting on the severity of the disorder [539]. Enalapril, a widespread antihypertensive drug, also possesses antioxidant features [540,541] and in vitro, displayed decreased cell proliferation and reduced hyaluronic acid levels in both the orbital fibroblasts affected by GO and the control fibroblasts [542]. A dedicated trial confirmed encouraging preclinical results in 12 patients with mild GO treated with enalapril for 6 months, concluding a beneficial impact on the clinical course and disease progression [543].

After reviewing existing publications on this theme, it appears reasonable that antioxidants may be considered a promising additional therapy in combination with corticosteroids to improve the quality of life of patients and slow down disease progression, as well as reduce the severity of the inflammation in GO [521].

### 5.3. Infiltrative Optic Neuropathies

Infiltrative processes of the optic nerve likewise generate RGC degeneration. An example is the leukemic optic neuropathy, mainly associated with acute lymphoblastic leukemia [544,545,546]. The involvement of the optic nerve in leukemic cases remains, however, a rare circumstance, which may mostly appear in relapses and late in the disease course [547,548]. The optic nerve head affected by infiltration may dangerously mimic a disc edema, a pathological sign detected by a large variety of different optic neuropathies [549]. In a fundoscopic examination, ONH can show irregular and nodular conformations [549]. Visual impairment is typically due to the neoplastic infiltration of ONH, which causes nerve fibers and vascular compressions [550]. Diagnostically, an MRI scan of the brain is essential for early detection and to possibly rapidly begin an appropriate therapy [551]. Pathogenetically, in relation to the possible direct involvement of oxidative stress in the pathophysiology of these pathologies, the current literature presents no dedicated studies on the argument.

### 5.4. Congenital Anomalies of the Optic Nerve

Optic nerve congenital anomalies also constitute a subgroup of optic neuropathies and include, for example, optic nerve hypoplasia and optic disc colobomas [552,553], the latter with a reported prevalence in children of ~8.9 in 100,000 [554]. Visual impairment in this large group of disorders may appear isolated or as part of a systemic malformation syndrome [555]. Congenital visual defects or visual loss are commonly associated with this devastating class of disorders [555]. No studies were performed to analyze the pathogenetic roles of ROS in these diseases.

### 5.5. Nutritional and Toxic Optic Neuropathies

Nutritional deficits, such as in case of vitamin B12 (cyanocobalamin), B1 (thiamine), or B9 (folic acid) deficiency, as well as intoxications, caused, for example, by ethambutol, amiodarone, or antibiotics, such as chloramphenicol [556,557,558], may cause optic nerve disorders. The prevalence of these pathologies is variable and typically depends on social, economic, geographical, and historical factors [31,32]. Some examples are the optic neuropathy in prisoners of the Japanese during World War II [559], the Cuban epidemic optic neuropathy [560], and the Tanzanian epidemic optic neuropathy [561], all of which were caused by metabolic deficiencies. Interestingly, nutritional optic neuropathies are currently becoming increasingly common as a consequence of bariatric surgery as well as strict vegetarian and vegan diets, thereby awaking scientific interest [32]. Nutritional optic neuropathies are usually caused by the deficiency of molecules which are crucial for the normal functionality of mitochondria, such as diverse subtypes of vitamin B [562]. Both intoxications or vitamin deficiencies usually manifest with optic disc pallor and anomalies in the papillomacular nerve fiber bundle and symptomatically through reduced color perception and detectable visual field defects, corresponding to the central or cecocentral scotomas, as observed in cases of LHON [556,562]. For example, toxic optic neuropathies caused by a chronic intake of chloramphenicol can mimic an acute stage of LHON [558]. Chloramphenicol can suppress mitochondrial protein synthesis, inducing an alteration in the mitochondrial structure and subsequently, dysfunction, with decreasing ATP and increasing ROS, which collectively reflect in LHON-like symptoms [563].

Possible treatments in these disorders obviously depend on the etiology. Indeed, nutritional optic neuropathies can be treated through vitamin B supplementations [564,565]. Toxic optic neuropathies should be treated rapidly after diagnosis through stopping the drug therapy in the case of pharmacological etiology and stopping smoking or alcohol consumption [557]. In this context, the dedicated studies also demonstrated the effectiveness of erythropoietin in improving visual acuities in patients affected by methanol-related toxic optic neuropathies [566,567].

## 6. Conclusions and Future Directions

Optic nerve disorders pose significant challenges due to their prevalence, severity, and economic implications for patients and the healthcare system. The development and proposal of new effective curative strategies are crucial in addressing these conditions. The field of antioxidative research has garnered increasing scientific interest, with numerous studies demonstrating promising preclinical results. Oxidative stress has emerged as a potential therapeutic target in animal models of glaucoma, LHON, AION, and ON. Clinical translations of these findings have already begun, as evidenced by the licensing of idebenone for LHON. However, considering the relative novelty of these potential antioxidants and the unknown aspects of their tolerability in humans, the careful planning of clinical studies is necessary. The high prevalence of glaucoma worldwide suggests that the cost-effective use of new antioxidant medications is feasible. Nonetheless, large-scale trials are imperative to ensure the applicability and long-term safety of these treatments for a large population of patients. The severity and rarity of conditions such as LHON may prompt exploration of new pharmacological avenues, including the antioxidant approach, as demonstrated by the development of idebenone [329]. Similarly, the lack of effective drugs for AION highlights the need to explore new directions, including the “antioxidant opportunity”.

Despite several positive findings in preclinical investigations, many potential antioxidant drugs have failed to progress beyond phase II in the corresponding clinical trials [568,569,570]. Currently, only a limited number of clinical studies are planned in the antioxidant field. Recent studies have examined the reasons for these failures, which include the low bioavailability of antioxidants, limited tissue targeting, poor antioxidant capacity, and high drug dosing that may be toxic for human use [571]. Additionally, some clinical investigations may have been limited by slow disease progression, such as in the case of glaucoma, resulting in long follow-up periods and a lack of sensitive biomarkers [568]. Significant efforts are currently being made to develop efficient drug delivery systems to enhance the translational success of potential antioxidants [445,446]. Alternative clinical trial designs, such as adaptive clinical trials, offer flexibility by allowing modifications based on predefined criteria [572,573].

In conclusion, the findings of our review have compiled significant preclinical evidence supporting the potential of targeting oxidative stress as a therapeutic approach in animal models of optic nerve diseases. These results highlight the importance of further advancing clinical trials in this research area, as they hold immense potential to bridge the current gap between preclinical and translational applications. By implementing and refining clinical trials, we can move closer to harnessing the benefits of oxidative stress modulation for the treatment of optic nerve diseases, ultimately improving patient outcomes and advancing therapeutic strategies in this field.

## Figures and Tables

**Figure 1 antioxidants-12-01465-f001:**
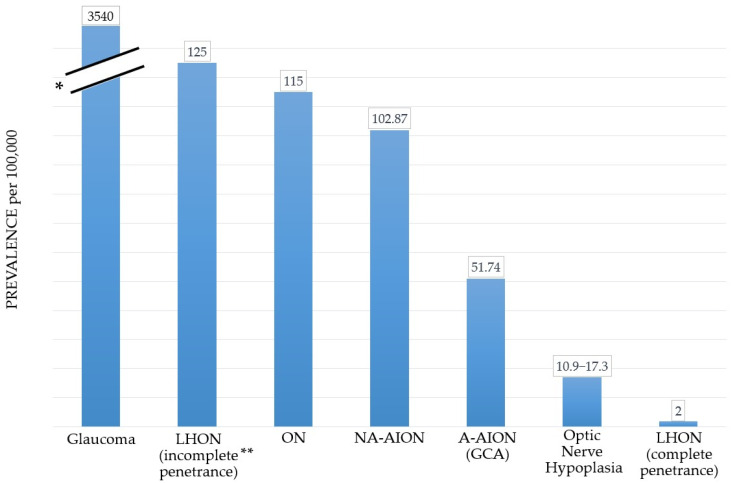
Prevalence (per 100,000) in some of the most frequent optic neuropathies. LHON: Leber’s hereditary optic neuropathy; ON: optic neuritis; NA-AION: nonarteritic anterior ischemic optic neuropathy; A-AION: arteritic anterior ischemic optic neuropathy; GCA: giant cell arteritis. * We used a *y*-axis break in consideration of the remarkably higher prevalence of glaucoma compared to all other optic nerve disorders. ** LHON prevalence in case of incomplete penetrance is meaningfully higher than in complete penetrance due to the high frequency of the variant mutant carriers.

**Figure 2 antioxidants-12-01465-f002:**
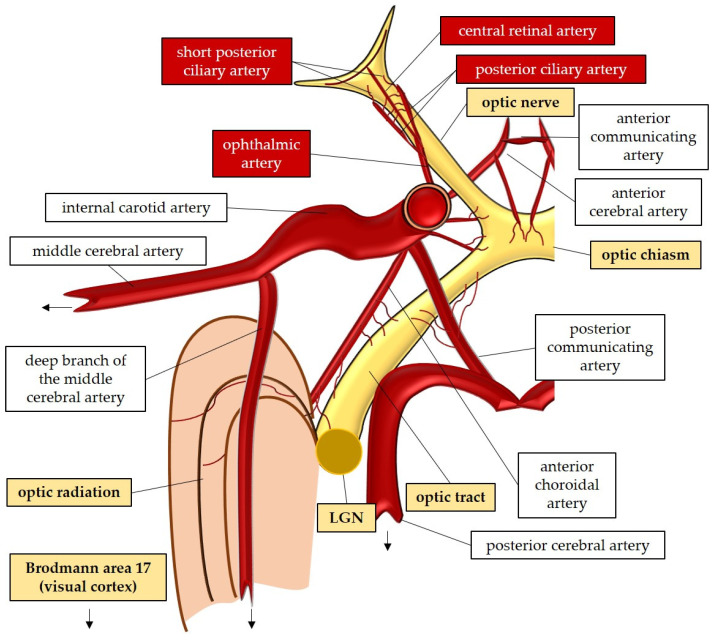
Anatomy and perfusion of the visual pathway. LGN: lateral geniculate nucleus.

**Figure 3 antioxidants-12-01465-f003:**
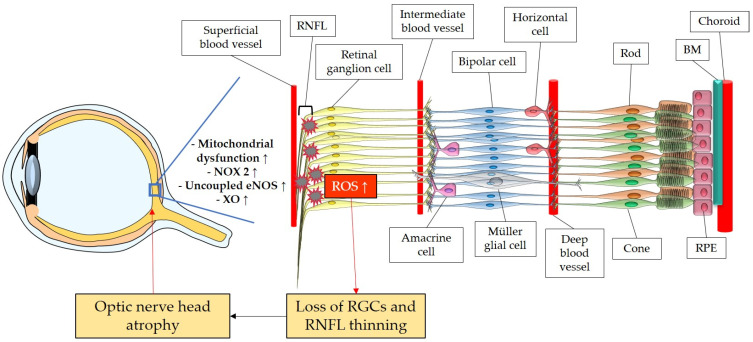
Model representing the ROS impact on the retina and on the optic nerve. ROS: reactive oxidative species; NOX2: NADPH oxidase type 2; XO: xanthine oxidase; eNOS: endothelial nitric oxide synthase; RGC: retinal ganglion cell; RNFL: retinal nerve fiber layer; RPE: retinal pigment epithelium; BM: Brunch’s membrane. Up arrows mean increase or upregulation.

**Figure 4 antioxidants-12-01465-f004:**
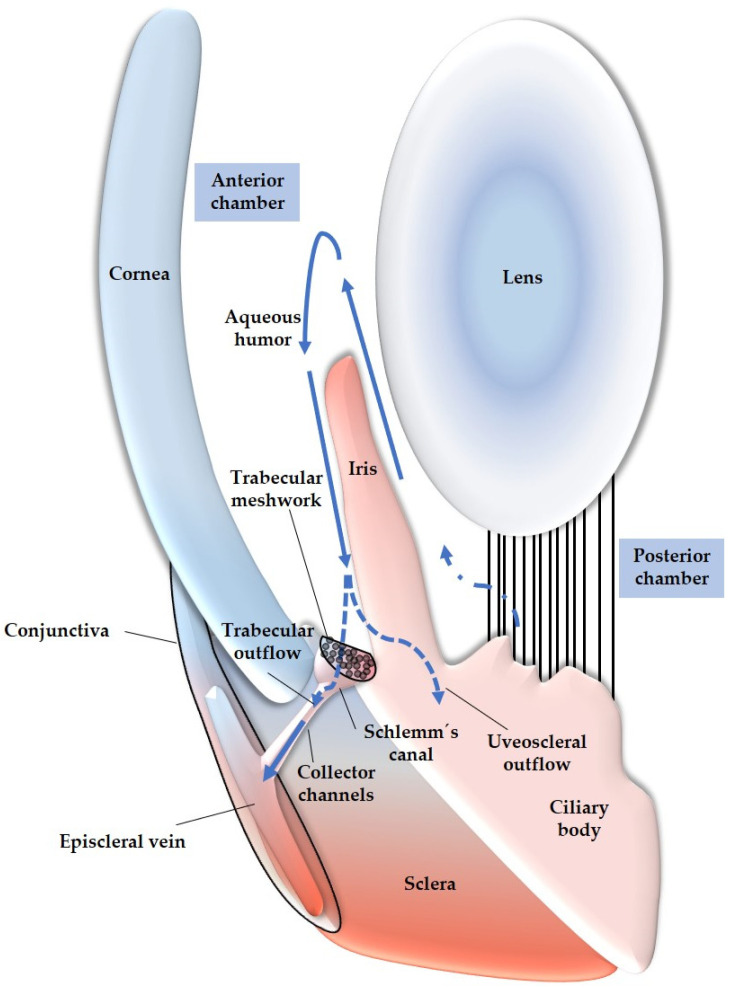
Anatomic overview of aqueous humor outflow, with focus on the drainage system through the trabecular meshwork and the Schlemm’s canal to the superficial veins.

**Figure 5 antioxidants-12-01465-f005:**
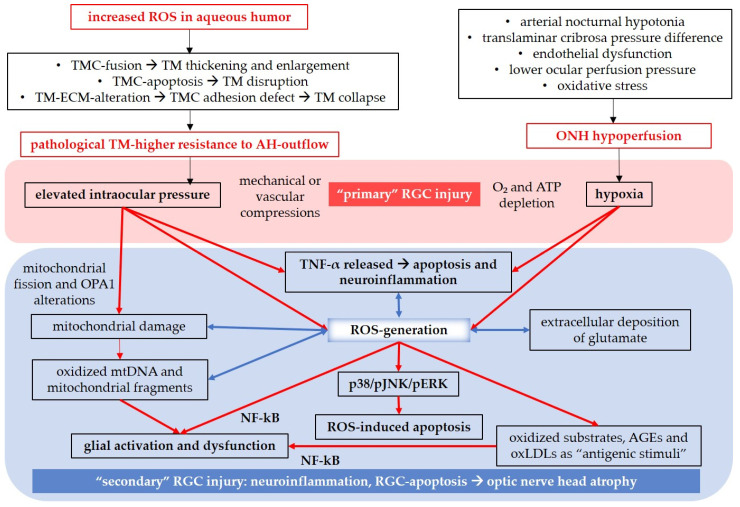
Model of etiopathogenesis in glaucomatous optic neuropathies. AH: aqueous humor; TM: trabecular meshwork; TMC: trabecular meshwork cell; ONH: optic nerve head; OPA1: optic atrophy 1 gene; TNF-α: tumor necrosis factor alpha; NF-kB: nuclear factor “kappa-light-chain-enhancer” of activated B-cells; ATP: adenosintriphosphat; RGC: retinal ganglion cell; oxLDL: oxidized low density lipoprotein; AGE: advanced glycation end product; pJNK: c-Jun N-terminal kinase; pERK: extracellular-signal-regulated kinase.

**Figure 6 antioxidants-12-01465-f006:**
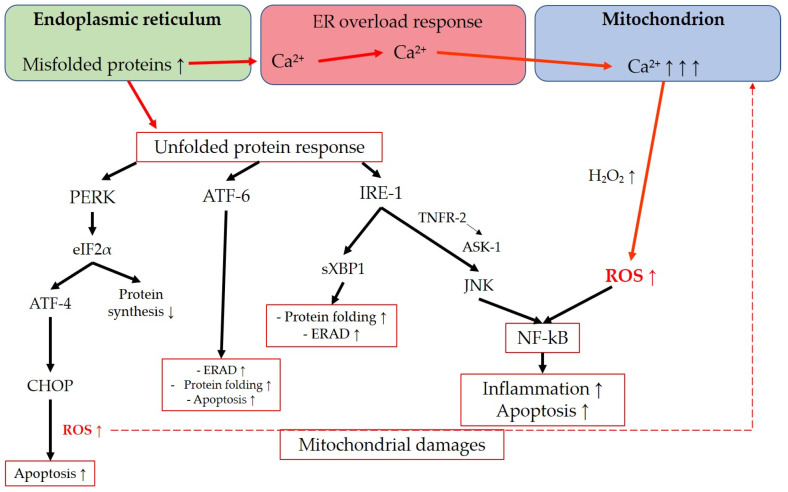
Role of calcium and reactive oxygen species in the interplay between ER and mitochondria. ER: endoplasmic reticulum; PERK: protein kinase RNA-like ER kinase; ATF: activating transcription factor; IRE-1: inositol-requiring protein 1; CHOP: CCAAT-enhancer-binding protein homologous protein; eIF2α: eukaryotic initiation factor 2α; sXBP1: spliced X-box binding protein-1; TNFR-2: tumor necrosis factor alpha receptor 2; ASK-1: apoptosis signal-regulating kinase 1; JNK: c-Jun N-terminal kinase; ERAD: ER-associated degradation; NF-kB: nuclear factor “kappa-light-chain-enhancer” of activated B-cells. Up arrows mean increase or upregulation. Down arrows mean decrease.

**Figure 7 antioxidants-12-01465-f007:**
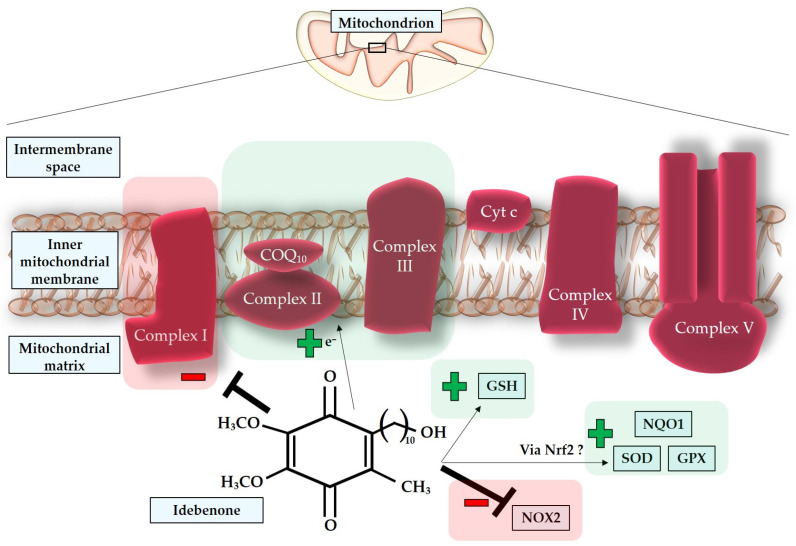
Direct and indirect effect of idebenone on the mitochondrial oxidative metabolism. COQ_10_: cofactor Q 10; Cyt c: cytochrome c; GSH: glutathione; SOD: superoxide dismutase; GPX: glutathione peroxidase; NQO1: NAD(P)H quinone oxidoreductase 1; NOX2: nicotinamide adenine dinucleotide phosphate oxidase 2; Nrf2: nuclear factor erythroid-derived 2-related factor 2.

**Figure 8 antioxidants-12-01465-f008:**
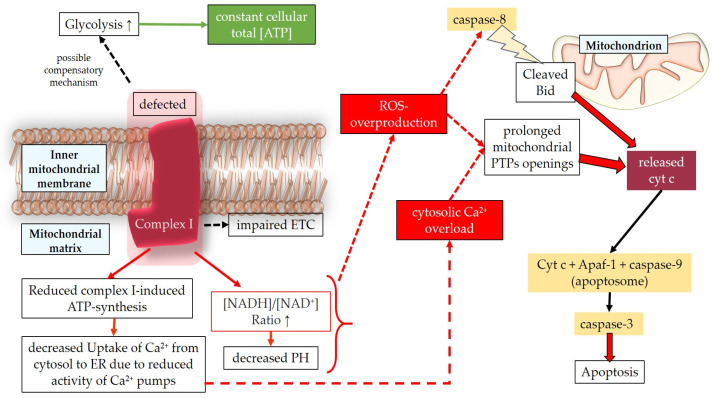
Model of LHON pathogenesis. NAD: nicotinamide dinucleotide; ETC: electron transport chain; ATP: adenosine triphosphate; cyt c: cytochrome c; Bid: BH3 interacting-domain death agonist; Apaf-1: apoptotic protease-activating factor 1; PTP: permeability transition pore.

## Data Availability

Not applicable.

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
