# Peer review of "Oxidative Stress: A Suitable Therapeutic Target for Optic Nerve Diseases?"

_antioxidants, 2023, doi:10.3390/antiox12071465_

Round 1

Reviewer 1 Report

This manuscript delves into optic nerve diseases, a spectrum of disorders that culminate in sight-threatening conditions due to the loss of retinal ganglion cells (RGCs) and optic nerve atrophy. It particularly emphasizes four extensively studied optic nerve diseases: glaucoma, Leber's hereditary optic neuropathy (LHON), anterior ischemic optic neuropathy (AION), and optic neuritis (ON). The paper underscores the pivotal role of oxidative stress, characterized by an overabundance of reactive oxygen species (ROS), in instigating cell death through apoptosis, autophagy, and inflammation in these diseases. Furthermore, it explores potential treatment methods such as natural compounds and nitric oxide inhibitors. The manuscript concludes that the field of antioxidant research has garnered increasing scientific interest, with many studies showing promising preclinical results.

The paper only focuses on four optic nerve diseases, and there may be other diseases not covered in this review. Lastly, while the paper discusses potential therapeutic effects against oxidative stress, it must be noted that further research is needed to fully understand the effectiveness and safety of such treatments.

In the treatment plans for traumatic or inflammatory neurodegenerative models, it is recommended discuss more promising treatments. For example, for models with complete RGC degeneration, non-invasive ultrasound stimulation may be a potential way to restore vision: https://doi.org/10.1109/TUFFC.2022.3220568.

Traditional implantable treatments include: https://doi.org/10.1038/s41467-022-31599-4. Literature on completely detached retinal optic nerve prostheses includes: https://doi.org/10.1038/s41551-019-0446-8.

As well as bypassing the visual pathway, directly stimulating the visual center's visual center prostheses literature: https://doi.org/10.3390/bioengineering10050577.

In addition, it is also recommended for the author(s) to incorporate additional pertinent references into the reference list to enhance the depth of research on neuromodulation and ultrasound-based biomedical neurodegeneration methods. https://doi.org/10.1152/ajpendo.00218.2019;

https://doi.org/10.1111/cns.14078.

https://doi.org/10.1016/j.nanoen.2021.106123

https://doi.org/10.1002/adfm.202200589

https://doi.org/10.1109/TBME.2022.3215498

https://doi.org/10.1109/TUFFC.2022.3190400

https://doi.org/10.1002/smll.202207600.

Author Response

We thank the reviewer for the comments and suggestions. According to these suggestions, we made changes in the text.

1.)        The paper only focuses on four optic nerve diseases, and there may be other diseases not covered in this review.

Response to 1.) We provide an in-depth description the four most prevalent optic neuropathies (glaucoma, Leber’s hereditary optic neuropathy, anterior ischemic optic neuropathy, optic neuritis) but we also, referring to the existing literature, characterized in detail the traumatic, dysthyroid, nutritional and toxic optic neuropathies, focusing on the role of oxidative stress in those disorders.

2.)        Lastly, while the paper discusses potential therapeutic effects against oxidative stress, it must be noted that further research is needed to fully understand the effectiveness and safety of such treatments.

Response to 2.) We added such statements, for example in TON. See our addition in text (5.1.3; lines 1290-1291, changes are underlined).

3.) In the treatment plans for traumatic or inflammatory neurodegenerative models, it is recommended discuss more promising treatments. For example, for models with complete RGC degeneration, non-invasive ultrasound stimulation may be a potential way to restore vision: https://doi.org/10.1109/TUFFC.2022.3220568

Traditional implantable treatments include:

https://doi.org/10.1038/s41467-022-31599-4

Literature on completely detached retinal optic nerve prostheses includes:

https://doi.org/10.1038/s41551-019-0446-8

As well as bypassing the visual pathway, directly stimulating the visual center's visual center prostheses literature: https://doi.org/10.3390/bioengineering10050577

In addition, it is also recommended for the author(s) to incorporate additional pertinent references into the reference list to enhance the depth of research on neuromodulation and ultrasound-based biomedical neurodegeneration methods.

https://doi.org/10.1152/ajpendo.00218.2019

https://doi.org/10.1111/cns.14078.

https://doi.org/10.1016/j.nanoen.2021.106123

https://doi.org/10.1002/adfm.202200589

https://doi.org/10.1109/TBME.2022.3215498

https://doi.org/10.1109/TUFFC.2022.3190400

https://doi.org/10.1002/smll.202207600.

Response to 3.) We are especially grateful for this comment. We now discuss neuro-modulation based on biomedical ultrasound stimulation for traumatic and inflammatory optic disorders (5.1.1; lines 2200-2221) and added the suggested references (changes in the text are underlined).

Reviewer 2 Report

Tthe manuscript by Buonfiglio et al. is intended to provide an overview on the role of ROS-related processes underlying four extensively studied optic nerve diseases. However, it tends to be particularly verbose.  The manuscript is far too long, I suggest to substantially shorten the descriptive parts, such as those on the mitochondrial structure and function and the long excursus on the cellular antioxidant s, leaving the details to the cited literature. 

The topic is worthy. The work is good, well written and structured. This reviewer is convinced that retinal and optic nerve diseases are primarily related to oxidative stress, which notably mainly comes from the mitochondrial respiratory chain, that the Authors correctly describe (in too much detail!).

However a flaw of the paper is the fact that Authors, who consider their work accurate and comprehensive in describing all possible sources of ROS in the nervous system  and optic nerve completely disregarded a number of experimental papers that show the existence of a primary source of ROS in the myelin sheath, in particular of the optic nerve myelin. Out of 572 cited references it is unacceptable that none reported the paper describing a possible source of ROS embedded inside the optic nerve sheath: i.e.  the whole  respiratory chain. Authors must cite and comment the literature on this aspect (doi: 10.1007/s11064-015-1712-0). Myelin has a trophic role for the axon to whihc it supplies nutrients. Recently it has been stated that “Improving … myelin integrity could be a promising target to delay development and slow progression of AD.” (doi.org/10.1038/s41586-023-06120-6 ). This may be true also for other neurodegerative diseases.

Furthermore, about  LHON, the Authors did not take into account the news that  the  treatment to which specifically References 338 and 339 refer,  has now been withdrawn (https://www.ema.europa.eu/en/medicines/human/withdrawn-applications/lumevoq): Gensight Biologics SA,  on last 20 April 2023, withdrew its application for a marketing authorization of Lumevoq for treating loss of vision due to LHON, as it was found not effective. This must be commented upon.

Author Response

We thank the reviewer for the comments and suggestions. According to these suggestions, we made changes in the text.

1.)        The manuscript is far too long, I suggest to substantially shorten the descriptive parts, such as those on the mitochondrial structure and function and the long excursus on the cellular antioxidants, leaving the details to the cited literature.

Response to 1.) According to the reviewer’s suggestion, we shortened the passage under subheader 3.1 in the descriptive part regarding the structure and function of mitochondria as well as under subheader 3.2 on the cellular antioxidants. Please see our changes in the text (lines 163-220, changed passages are underlined).

2.)        However, a flaw of the paper is the fact that Authors, who consider their work accurate and comprehensive in describing all possible sources of ROS in the nervous system and optic nerve completely disregarded a number of experimental papers

that show the existence of a primary source of ROS in the myelin sheath, in particular of the optic nerve myelin. Out of 572 cited references it is unacceptable that none reported the paper describing a possible source of ROS embedded inside the optic nerve sheath: i.e. the whole respiratory chain. Authors must cite and comment the literature on this aspect (doi: 10.1007/s11064-015-1712-0). Myelin has a trophic role for the axon to which it supplies nutrients. Recently it has been stated that “Improving …myelin integrity could be a promising target to delay development and slow progression of AD.”(doi.org/10.1038/s41586-023-06120-6). This may be true also for other neurodegenerative diseases.

Response to 2.) According to the reviewer’s suggestion, we added a passage on the link between myelin an ROS in chapter 2 (lines 144-160).

3.) Furthermore, about LHON, the Authors did not take into account the news that the treatment to which specifically References 338 and 339 refer, has now been withdrawn (https://www.ema.europa.eu/en/medicines/human/withdrawn-pplications/lumevoq): Gensight Biologics SA, on last 20 April 2023, withdrew its application for a marketing authorization of Lumevoq for treating loss of vision due to LHON, as it was found not effective. This must be commented upon.

Response to 3.) We are very grateful to the Reviewer about these remarks. We added a paragraph on Lumevoq under subheader 4.3.1. (lines795-807) and mention the fact that it has been withdrawn by EMA.